# FAST AND ACCURATE LANGUAGE MODEL DECODING VIA PARALLEL TOKEN PROCESSING

## ABSTRACT

Autoregressive decoding suffers from an inherent efficiency bottleneck due to its sequential token generation process, where each token must be generated before the next can be processed. This sequential dependency significantly limits the ability to fully exploit the parallel processing power of modern hardware. While speculative decoding and layer skipping offer promising speedups, both approaches come with drawbacks. Speculative decoding relies on a secondary small "drafter" model, which not only increases memory overhead but may also be unavailable in many cases—the drafter must share the same tokenizer and vocabulary as the main model for compatibility between generated and verified tokens. Layer skipping, on the other hand, can cause discrepancies in the generated output compared to standard autoregressive decoding, as skipped layers do not compute the key-value (KV) cache that plays a crucial role in predicting future tokens. In this work, we introduce a fast and accurate decoding method, ParaDecode, which accelerates autoregressive decoding while ensuring output parity, without the need for auxiliary models or changes to original model parameters. Our approach is driven by the observation that many tokens—particularly simple or highly-predictable ones—can be accurately predicted using intermediate layer representations, without requiring computation through the entire model. Once the model reaches a certain confidence, further layers are unlikely to significantly alter the prediction. ParaDecode generates tokens at an intermediate layer when confidence is sufficiently high. This allows the next token computation to start immediately, in parallel with the completion of the KV cache computation for the early-predicted token in its remaining layers. This parallelism, implemented using batched matrix operations, enables simultaneous processing of multiple tokens across different layers, thereby maximizing hardware utilization and reducing overall decoding latency. To ensure output consistency, a final verification step is applied to guarantee that the early-predicted tokens match the results of standard autoregressive decoding. Experiments across diverse generation tasks, including text summarization, code generation, and mathematical reasoning, demonstrate that ParaDecode consistently achieves superior decoding throughput compared to baselines with up to **1.53×** speedup, while guaranteeing output parity with standard autoregressive decoding.[1]

## 1 INTRODUCTION

The autoregressive decoding process in large language models (LLMs) is increasingly becoming a critical efficiency bottleneck for text generation (Khoshnoodi et al., 2024). As each token generation depends on previously generated ones, the inherently sequential nature of this process severely limits parallelization capabilities on modern hardware (Miao et al., 2023a). This challenge is becoming more pressing due to two key trends. Firstly, LLMs continue to grow exponentially in size (Kaplan et al., 2020; Hoffmann et al., 2022; DeepSeek, 2024), with billions or even trillions of parameters (Achiam et al., 2023; Zeng et al., 2023; Dubey et al., 2024; Jiang et al., 2024; Anthropic, 2024), resulting in substantially more time-consuming and resource-intensive computations at each step of token generation. Secondly, model-generated outputs are becoming progressively longer, driven by emerging applications such as prolonged chain-of-thought reasoning (OpenAI, 2024; Snell et al., 2024; Brown et al., 2024) and long-form content creation (Pham et al., 2024; Bai et al., 2024), which

---

[1]Code is included in the submitted supplementary material.

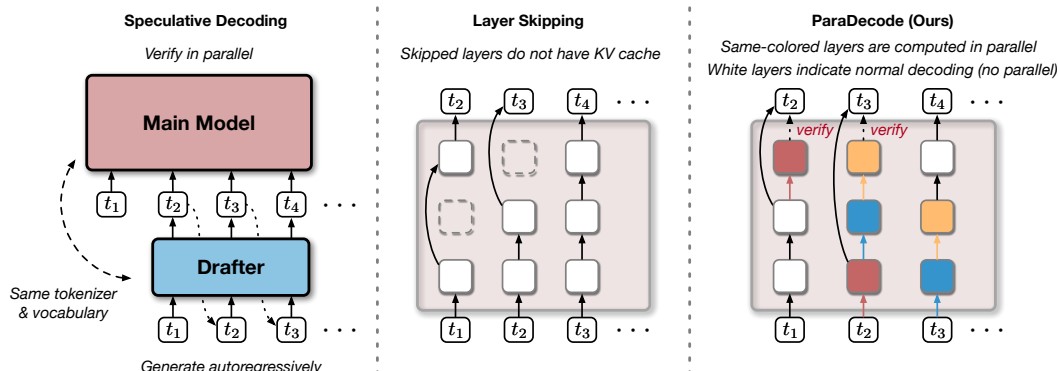

Figure 1: (Left) Speculative decoding relies on an auxiliary drafter model, leading to increased memory usage and requiring the same tokenizer and vocabulary as the main model. (Middle) Layer skipping bypasses certain layers, which results in missing KV cache at those layers and can introduce discrepancies in future token predictions. (Right) ParaDecode (our proposed method) accelerates decoding by processing tokens in parallel. When an intermediate layer confidently predicts the current token, we begin computing the next token position in parallel with the remaining layers' KV cache. A final verification step ensures output consistency with standard autoregressive decoding.

require massive inference steps. The confluence of these factors leads to significant latency in text generation, amplifying the urgent need for more efficient decoding methods.

To accelerate autoregressive decoding, two primary approaches have emerged: speculative decoding and layer skipping, as illustrated in Figure 1. Speculative decoding (Leviathan et al., 2023; Chen et al., 2023; Liu et al., 2023; Miao et al., 2023b; He et al., 2024; Huang et al., 2024) employs a lightweight secondary model, called a "drafter," to generate candidate tokens at lower latency, which are then verified in parallel by the larger main model. However, the reliance on a separate drafter model increases memory overhead and can be impractical in many cases, since the drafter must share the same tokenizer and vocabulary as the main model to ensure token compatibility. Layer skipping (Huang et al., 2018a; Elbayad et al., 2020; Elhoushi et al., 2024; Del Corro et al., 2023; Raposo et al., 2024; Geva et al., 2022; Din et al., 2023), in contrast, reduces computation cost by selectively bypassing certain layers during token generation. This approach often requires designing new model architectures and intricate training methods (Elhoushi et al., 2024; Raposo et al., 2024). Although effective at reducing latency, layer skipping often leads to discrepancies in output quality compared to standard autoregressive decoding (Schuster et al., 2022; Liu et al., 2024a). Specifically, skipped layers do not compute the key-value (KV) cache, which is essential for maintaining consistency in the model's predictions of future tokens. As a result, while both speculative decoding and layer skipping provide promising speedups, they come with trade-offs that pose challenges for their widespread adoption in practice.

In this work, we propose ParaDecode, a fast and accurate decoding method designed to accelerate autoregressive decoding through parallel token processing. ParaDecode builds on the insight that many simple and predictable tokens can be accurately generated at intermediate layers, without requiring a full pass through all model layers (Schuster et al., 2022). To optimize token generation quality at intermediate layers, we introduce lightweight language model (LM) heads at intermediate layers, which are trained to minimize the KL divergence between their predictions and those of the final layer, while keeping the original model parameters frozen. Our preliminary studies show that when predictions at intermediate layers are sufficiently confident, subsequent layers are unlikely to significantly alter the output. Based on this observation, ParaDecode generates tokens from intermediate layers when confidence is high, and simultaneously initiates processing of the next token in parallel with the remaining layers' KV cache computation for the current token. This parallelism is achieved through batched matrix operations, allowing multiple tokens to be processed across different layers simultaneously, maximizing hardware utilization and improving overall decoding throughput. Once the KV cache for all layers is computed, we verify that the early-predicted token matches the standard autoregressive decoding result, ensuring consistency. Compared to speculative decoding and layer skipping, ParaDecode accelerates autoregressive decoding while maintaining output parity, without requiring auxiliary models or modifications to the original model parameters.

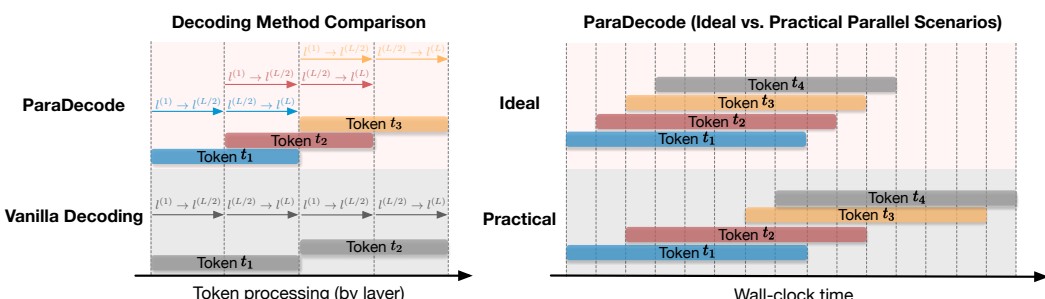

Figure 2: (Left): Vanilla autoregressive decoding processes tokens strictly in sequence, limiting opportunities for parallelization. In contrast, ParaDecode starts processing the first few layers of the next token in parallel with the remaining layers of the current token, once the model can confidently predict the next token using intermediate LM heads. For illustration purposes, early predictions are assumed to occur at layer $L/2$, though in practice, predictions can happen at other layers based on confidence. (Right): The ideal case is illustrated in the upper part, where all tokens are predicted early at shallow layers, leading to maximal parallelization. The lower diagram shows a more typical scenario, where tokens are early predicted at different intermediate layers, resulting in varying degrees of parallelization.

Our contributions are as follows: (1) We propose an intermediate-layer LM head training approach that optimizes token prediction at earlier layers and results in high-confidence early-predicted tokens, without modifying the original model parameters; (2) We introduce a parallel token processing technique that exploits batched matrix operations to concurrently process multiple tokens across different model layers, significantly improving hardware utility and decoding speed; and (3) ParaDecode demonstrates superior decoding throughput compared to existing methods across a variety of challenging text generation tasks with up to **1.53×** speedup, while maintaining output consistency with standard autoregressive decoding.

## 2 METHOD: PARADECODE

In this section, we present our proposed method ParaDecode, as illustrated in Figure 2. The core concept is to start processing the initial layers of subsequent tokens in parallel with completing the current token's layers, thereby enhancing overall decoding throughput via parallel computation. We introduce the techniques for enabling early predictions using intermediate layer representations in Section 2.1, followed by a detailed explanation of our parallel processing approach in Section 2.2.

### 2.1 LIGHTWEIGHT LANGUAGE MODEL HEADS ENABLE EARLY PREDICTIONS

**Off-the-shelf LMs struggle with early predictions.** Many tokens in natural language, such as stopwords and common phrase completions, can be easily predicted and do not require the full capacity of a model for accurate generation. However, off-the-shelf LMs typically have difficulty utilizing intermediate layers for next-token prediction, as the final-layer LM head is not trained to work with intermediate-layer representations. As shown in Figure 3a, applying the final-layer LM head to intermediate layers results in mostly low predicted probabilities for the tokens generated by the model, making early predictions with standard LMs challenging. Hence, prior research on early exiting often involves designing specific model architectures or fine-tuning existing models to enable intermediate-layer predictions (Elhoushi et al., 2024; Din et al., 2023; Del Corro et al., 2023). As a result, these acceleration methods typically fail to produce outputs consistent with those of the original off-the-shelf LMs due to changes in architecture and parameters.

**Training intermediate-layer LM heads with original model parameters frozen.** We hypothesize that many intermediate-layer representations may already contain sufficient information for predicting the next token, but the original LM head cannot directly harness their potential. To facilitate early predictions using intermediate-layer representations without further fine-tuning them, we introduce trainable LM heads $\boldsymbol{\theta}^{(i)} = \{\boldsymbol{e}_t^{(i)}\}_{t \in \mathcal{V}}$ at each candidate early prediction layer $l^{(i)}$. They take the hidden representations $\boldsymbol{h}^{(i)}$ at layer $l^{(i)}$ as frozen features and predict the next word distribution $p_{\boldsymbol{\theta}^{(i)}}(t|\boldsymbol{h}^{(i)})$, which are trained to approximate the last-layer prediction $p^*(t|\boldsymbol{h}^*)$ ($\boldsymbol{h}^*$ is the last-layer

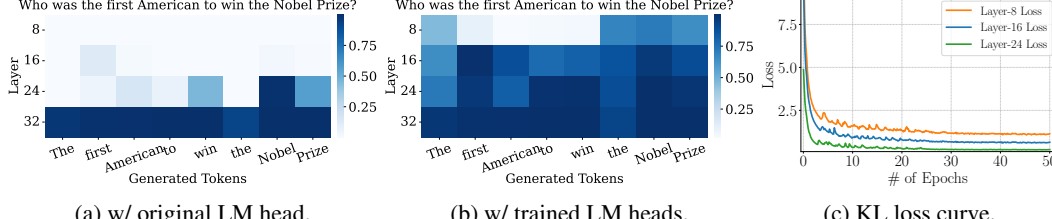

(a) w/ original LM head.        (b) w/ trained LM heads.        (c) KL loss curve.

Figure 3: Probabilities of model-generated tokens predicted at the 8th, 16th, 24th, and 32nd (final) layers of the fine-tuned Llama-3.1-8B-Instruct are shown using (a) the original last-layer language model (LM) head and (b) our newly introduced lightweight LM heads. These new LM heads are trained to minimize the KL divergence loss relative to the final layer predictions, while keeping all original model parameters frozen. The LM heads enable close approximation of the final predictions, as seen in (b), where many tokens have confident predictions at intermediate layers, and in (c), where the KL divergence loss relative to the final layer predictions is minimal.

hidden states) by minimizing the following KL divergence loss:

$$p_{\boldsymbol{\theta}^{(i)}}(t|\boldsymbol{h}^{(i)}) = \frac{\exp\left(\boldsymbol{e}_t^{(i)} \cdot \boldsymbol{h}^{(i)}\right)}{\sum_{t' \in \mathcal{V}} \exp\left(\boldsymbol{e}_{t'}^{(i)} \cdot \boldsymbol{h}^{(i)}\right)}, \quad \mathcal{L}(\boldsymbol{\theta}^{(i)}) = \mathrm{KL}\left(p^*(t|\boldsymbol{h}^*)\middle\|p_{\boldsymbol{\theta}^{(i)}}(t|\boldsymbol{h}^{(i)})\right).$$

Since we do not update $\boldsymbol{h}^{(i)}$, the original model parameters remain unchanged, and only the newly added LM heads are trained. As demonstrated in Figure 3c, training these intermediate-layer LM heads results in good approximations of the last-layer outputs, as indicated by the low KL divergence loss at the end of training. This supports our hypothesis that intermediate-layer representations contain ample information for predicting the next token, and simple transformations via new LM heads can effectively extract this information. Consequently, with our trained LM heads, many tokens' predicted probabilities are notably high at intermediate layers, as shown in Figure 3b.

**Lightweight LM head implementation.** The LM heads $\boldsymbol{\theta}^{(i)} = \{\boldsymbol{e}_t^{(i)}\}_{t \in \mathcal{V}}$ are typically represented by a weight matrix $\boldsymbol{E}^{(i)} \in \mathbb{R}^{|\mathcal{V}| \times d}$ where $d$ is the model dimension. Given the large vocabulary size of LMs, learning a separate LM head for each early prediction layer leads to a substantial increase in the number of parameters. Based on the observation that the weight matrix $\boldsymbol{E}^{(i)}$ is always applied to the hidden states $\boldsymbol{h}^{(i)}$ to compute the probability over the vocabulary $p_{\boldsymbol{\theta}^{(i)}} = \mathrm{Softmax}(\boldsymbol{h}^{(i)}\boldsymbol{E}^{(i)\top})$, to reduce the parameter cost, we decompose it as $\boldsymbol{E}^{(i)} = \boldsymbol{E}^*\boldsymbol{T}^{(i)}$ where $\boldsymbol{E}^*$ is the last-layer LM head weights, and $\boldsymbol{T}^{(i)} \in \mathbb{R}^{d \times d}$ is a learnable transformation matrix. As $d \ll |\mathcal{V}|$ for most LLMs, learning $\boldsymbol{T}^{(i)}$ is much more parameter-efficient than learning $\boldsymbol{E}^{(i)}$ directly. We defer the proof that learning $\boldsymbol{T}^{(i)}$ retains the full expressiveness of learning $\boldsymbol{E}^{(i)}$ to Appendix A. To further reduce the number of trainable parameters, low-rank adaptation methods (Hu et al., 2022) can be explored, which we leave for future work.

In practice, in an Llama3.1-8B-Instruct model (Dubey et al., 2024), each lightweight LM head $\boldsymbol{T}^{(i)}$ introduces only 16M parameters ($d = 4096$), while a full LM head $\boldsymbol{E}^{(i)}$ would require 0.5B parameters ($|\mathcal{V}| = 128K$). We find that a small-scale corpus with around 15K samples is sufficient to train these lightweight LM heads to achieve good generalization.

## 2.2 PARALLEL TOKEN PROCESSING VIA BATCHED MATRIX OPERATIONS

**Early predictions trigger parallel processing.** As shown in Figure 3b, when a token's predicted probability is sufficiently high with intermediate LM heads (introduced in Section 2.1), subsequent layers are unlikely to change the predictions significantly. Based on this observation, we generate the next token $t$ at layer $l^{(i)}$ when its probability surpasses a predefined threshold:

$$t \sim p_{\boldsymbol{\theta}^{(i)}}(t'|\boldsymbol{h}^{(i)}) \quad \text{and} \quad p_{\boldsymbol{\theta}^{(i)}}(t|\boldsymbol{h}^{(i)}) > \gamma, \tag{1}$$

where $\gamma$ is a hyperparameter, and any sampling strategy can be employed (*e.g.*, greedy or nucleus sampling (Holtzman et al., 2020)). This allows parallel processing of the next token while completing the current token's processing. Notably, it is necessary to finish processing the remaining layers

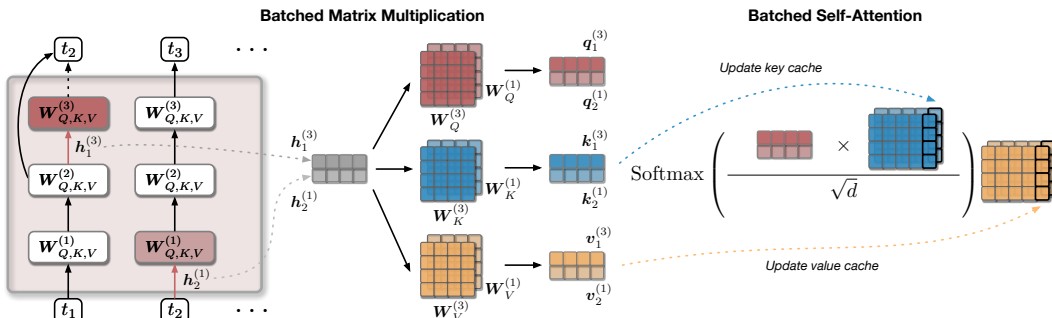

Figure 4: Parallel token processing via batched operations illustrated for the self-attention module, with other Transformer modules (*e.g.*, feedforward networks) operating similarly. If token $t_2$ can be generated at layer 2 while processing $t_1$, its computation at layer 1 starts immediately. This enables the computation of layer 1 at $t_2$ to run concurrently with layer 3 at $t_1$. Batched matrix multiplication facilitates simultaneous calculations of query, key, and value states across the parallel layers. The self-attention calculation also utilizes batched operations with updated KV cache.

of the current token to obtain their KV cache—omitting this step would result in a missing KV cache at deeper layers, which would cause inconsistencies when computing future token representations.

**Batched matrix operations for parallelization.** Modern deep learning libraries (*e.g.*, PyTorch (Paszke et al., 2019)) offer various approaches for parallel computation, including pipeline parallelism (Huang et al., 2018b), multi-threading (Dean et al., 2012), and asynchronous execution (Zheng et al., 2017). However, our preliminary studies indicate that the actual speedups achieved by these parallelization techniques can vary widely. In many cases, these techniques do not enhance throughput and may even result in longer overall decoding time, likely due to the overhead associated with scheduling, managing, and synchronizing threads and processes. Therefore, we reformulate all parallel computations as batched matrix operations to optimize hardware utilization, taking advantage of GPUs' specialization, yielding the best practical throughput.

Figure 4 demonstrates a specific parallel processing example: when $t_2$ is generated at layer 2 of $t_1$'s processing, layer 3 computations for $t_1$ are executed concurrently with layer 1 computations for $t_2$. More generally, consider $N$ tokens $t_1, t_2, \ldots, t_N$ being processed in parallel at layers $l^{(t_1)}, l^{(t_2)}, \ldots, l^{(t_N)}$ ($l^{(t_1)} > l^{(t_2)} > \cdots > l^{(t_N)}$). The corresponding input hidden states for these layers are $\boldsymbol{h}^{(t_1)}, \boldsymbol{h}^{(t_2)}, \ldots, \boldsymbol{h}^{(t_N)}$. By stacking these input hidden states, we form a batched input $\boldsymbol{H}^{\oplus} = [\boldsymbol{h}^{(t_1)}; \boldsymbol{h}^{(t_2)}; \ldots; \boldsymbol{h}^{(t_N)}] \in \mathbb{R}^{N \times d}$. Similarly, we prepare the batched weight matrices $\boldsymbol{W}_X^{\oplus} = [\boldsymbol{W}_X^{(t_1)}; \boldsymbol{W}_X^{(t_2)}; \ldots; \boldsymbol{W}_X^{(t_N)}] \in \mathbb{R}^{N \times d \times d}$ where $X$ may represent $Q$, $K$, or $V$. For simplicity, we assume single-head attention with $Q, K, V \in \mathbb{R}^{d \times d}$, though this approach easily generalizes to multi-head attention. The batched matrix multiplication, denoted by $\odot$ and implemented using the `torch.matmul` function, is expressed as:

$$\boldsymbol{X}^{\oplus} = \boldsymbol{H}^{\oplus} \odot \boldsymbol{W}_X^{\oplus} = \left[ \boldsymbol{h}^{(t_1)} \boldsymbol{W}_X^{(t_1)}; \boldsymbol{h}^{(t_2)} \boldsymbol{W}_X^{(t_2)} \ldots; \boldsymbol{h}^{(t_N)} \boldsymbol{W}_X^{(t_N)} \right], \quad \boldsymbol{X} \in \{\boldsymbol{Q}, \boldsymbol{K}, \boldsymbol{V}\}.$$

We can then compute the self-attention outputs $\boldsymbol{O}^{\oplus}$ as follows:

$$\boldsymbol{O}^{\oplus} = \text{Softmax}\left( \frac{\boldsymbol{Q}^{\oplus} \odot \boldsymbol{K}_{\text{cache}}^{\oplus \top}}{\sqrt{d}} \right) \odot \boldsymbol{V}_{\text{cache}}^{\oplus},$$

where $\boldsymbol{K}_{\text{cache}}^{(t_i)}$ and $\boldsymbol{V}_{\text{cache}}^{(t_i)}$ denote the updated KV cache at token position $t_i$.

Other Transformer components, such as feedforward networks and LayerNorm (Ba et al., 2016), can also be expressed as batched matrix operations and parallelized in a similar fashion.

**Early prediction verification.** Regardless of the threshold hyperparameter $\gamma$ set in Equation (1), there is always a possibility that the early predicted token from intermediate layers differs from the final prediction. To ensure consistency with standard autoregressive decoding, we introduce a verification step for every early prediction once the KV cache for all remaining layers has been fully computed. Specifically, we compare the early predicted token $t$, sampled from the intermediate layer's distribution $p_{\boldsymbol{\theta}^{(i)}}(t'|\boldsymbol{h}^{(i)})$, with the gold predicted token $t^*$, sampled from the final layer's

distribution $p^*(t'|\boldsymbol{h}^*)$. If the two tokens do not match, we reject the early prediction:

$$\text{Reject } t \text{ if } t \neq t^*, \quad \text{where} \quad t \sim p_{\boldsymbol{\theta}^{(i)}}(t'|\boldsymbol{h}^{(i)}), \quad t^* \sim p^*(t'|\boldsymbol{h}^*).$$

When an early prediction $t$ is rejected, we halt the generation process, replace $t$ with the actual token $t^*$, and resume generation from $t^*$ onward. Although rejecting early predictions can lead to some wasted computation, we observe in practice that this happends rarely. For example, with $\gamma = 0.8$, only about 5% of all early predictions are rejected, which supports our observation that high-confidence predictions from intermediate layers tend to align with the final predictions.

**Overall algorithm** of ParaDecode is presented in Algorithm 1.

---

**Algorithm 1:** ParaDecode

---

**Input:** $\boldsymbol{x} = [x_1, x_2, \ldots, x_M]$: user prompt; $\mathcal{S} = \{l^{(i)}\}\big|_{i=1}^{|\mathcal{S}|}$: early prediction layers
**Parameter:** $\gamma$: early prediction threshold
**Output:** Generated output sequence $\boldsymbol{y} = [t_0, t_1, t_2, \ldots, t_N]$
KV cache $\leftarrow$ LM($\boldsymbol{x}$)                              // initialize KV cache by processing user prompt
$\boldsymbol{y} \leftarrow [t_0]$                              // initialize output sequence with BOS token
$\mathcal{P} \leftarrow \{\ \}$                  // initialize a dictionary to track (token, layer) pairs being processed in parallel
$\mathcal{P}[t_0] \leftarrow 0$                              // start processing BOS token at layer 0
**while** $\boldsymbol{y}[-1] \neq$ EOS                              // terminate generation upon generating EOS token
**do**
  $\quad$ update KV cache $\leftarrow$ LM($\boldsymbol{y}$; KV cache) with parallel processing of all layers in $\mathcal{P}$
  $\quad \mathcal{P}[t] \leftarrow \mathcal{P}[t] + 1, \forall t \in \mathcal{P}$                  // proceed to the next layer for all parallel layers
  $\quad$ **if** $l^{(i)} = \mathcal{P}[\boldsymbol{y}[-1]] \in \mathcal{S}$          // if the latest token $\boldsymbol{y}[-1]$ reaches an early prediction layer
  $\quad$ **then**
    $\quad\quad t \sim p_{\boldsymbol{\theta}^{(i)}}(t'|\boldsymbol{h}^{(i)})$                              // sample from the intermediate LM head
    $\quad\quad$ **if** $p_{\boldsymbol{\theta}^{(i)}}(t|\boldsymbol{h}^{(i)}) > \gamma$                  // if the probability surpasses the threshold
    $\quad\quad$ **then**
      $\quad\quad\quad \boldsymbol{y} \leftarrow \boldsymbol{y} \oplus t$                              // append token $t$ to output sequence
      $\quad\quad\quad \mathcal{P}[t] \leftarrow 0$                              // add token $t$ at layer 0 to parallel processing
  $\quad$ **if** $\exists t \in \mathcal{P}, \mathcal{P}[t] = L$                  // if any token being processed reaches the final layer $L$
  $\quad$ **then**
    $\quad\quad t^* \sim p^*(t'|\boldsymbol{h}^*)$                              // sample the gold token $t^*$ from final-layer predictions
    $\quad\quad$ **if** $t^* \notin \mathcal{P}$                              // verify early prediction if made previously
    $\quad\quad$ **then**
      $\quad\quad\quad$ Empty $\mathcal{P}$; $\mathcal{P}[t^*] \leftarrow 0$  // reject early prediction, halt generation, and start from $t^*$ at layer 0
    $\quad\quad$ remove $t$ from $\mathcal{P}$
**return** $\boldsymbol{y}$

---

## 3 EXPERIMENTAL SETUPS

### 3.1 EVALUATION TASKS AND BASELINES

**Evaluation tasks.** We evaluate our method on a diverse set of text generation tasks, including text summarization (*i.e.*, XSum (Narayan et al., 2018)), code generation (*i.e.*, HumanEval (Chen et al., 2021)), and mathematical reasoning (*i.e.*, GSM8K (Cobbe et al., 2021)), covering a broad spectrum of language model capabilities. Due to the page limit, please refer to Appendix B for a detailed introduction of the benchmarks.

**Baselines.** In our work, we primarily focus on comparing with efficient decoding baselines that provide output parity guarantees with standard autoregressive decoding techniques, such as speculative decoding (Leviathan et al., 2023; Chen et al., 2023) and self-speculative decoding (Zhang et al., 2024a). Despite their conceptual advantages, these methods have practical limitations. They often come with inherent constraints regarding model selection or necessitate task-specific model architectures. Such requirements significantly compromise their practical utility and present difficulties for real-world application—as we will demonstrate later (§ 4.3), they can only lead to quite limited speedup or even negative speedup results compared to standard decoding unless careful hyperparameter tuning is performed. Below, we provide a detailed introduction of the baselines.

Table 1: Speedup comparison between ParaDecode and baseline decoding methods across three tasks, including text summarization, code generation, and mathematical reasoning. All compared methods ensure generation parity with vanilla autoregressive decoding, and the speedup results are computed relative to the benchmarks established on the vanilla setup (*i.e.*, speedup = $1\times$ for vanilla autoregressive decoding). '–' represents not applicable, as there is no smaller drafter from the same family of the verifier model. The best performance is highlighted in ***bold***.

| Method | Text Summarization (XSum) | Code Generation (HumanEval) | Mathematical Reasoning (GSM8K) |
|---|---|---|---|
| SpecDecode (Leviathan et al., 2023) | | | |
| Llama3.1-8B$_{\text{INST}}$ | | | |
| no smaller drafter available | – | – | – |
| CodeLlama-34B$_{\text{INST}}$ | | | |
| w/ drafter CodeLlama-7B$_{\text{INST}}$ | $1.08\times$ | $1.41\times$ | $1.26\times$ |
| w/ drafter CodeLlama-13B$_{\text{INST}}$ | $1.18\times$ | $1.23\times$ | $1.10\times$ |
| Self-SpecDecode (Zhang et al., 2024a) | | | |
| Llama2-7B$_{\text{INST}}$ | $1.05\times$ | $1.09\times$ | $1.10\times$ |
| CodeLlama-13B$_{\text{INST}}$ | $1.03\times$ | $1.14\times$ | $1.12\times$ |
| CodeLlama-34B$_{\text{INST}}$ | $1.07\times$ | $1.14\times$ | $1.14\times$ |
| ParaDecode (Ours) | | | |
| Llama3.1-8B$_{\text{INST}}$ | $1.15\times$ | $1.31\times$ | $1.42\times$ |
| CodeLlama-34B$_{\text{INST}}$ | $\mathbf{1.24\times}$ | $\mathbf{1.45\times}$ | $\mathbf{1.53\times}$ |

**SpecDecode.** For this baseline, we consider two configurations: (1) CodeLlama-34B-Instruct as the main model (*i.e.*, verifier) with CodeLlama-13B-Instruct as the assistant model (*i.e.*, drafter), and (2) CodeLlama-34B-Instruct as the verifier with CodeLlama-7B-Instruct as the drafter. Note that it is impractical to evaluate SpecDecode with state-of-the-art moderate-sized models such as Llama3.1-8B-Instruct due to the absence of a smaller drafter model within the same family.

**Self-SpecDecode.** For a fair comparison, we adopt three backbone models for Self-SpecDecode: (1) Llama2-7B-chat, (2) CodeLlama-13B-Instruct, and (3) CodeLlama-34B-Instruct. Following Zhang et al. (2024a), we adopt an adaptive confidence threshold strategy and set the initial threshold $\gamma^0 = 0.6$, the max number of draft token $K = 12$, and the max number of generated token $T = 512$. We run the Bayesian optimization search for 200 iterations to determine skipped layers for configuring the drafter model. We use 4 instances randomly sampled from the training set of each task for the Bayesian optimization search as suggested in the original implementation. More implementation details regarding training and inference can be found in Appendix C.

## 4 RESULTS

### 4.1 MAIN RESULTS

**ParaDecode consistently achieves superior inference speedup across all benchmarks.** To validate the effectiveness of our method, we compare the proposed ParaDecode with state-of-the-art efficient decoding methods SpecDecode and Self-SpecDecode across a wide range of challenging text generation tasks. As presented in Table 1, our method consistently delivers superior speedup compared to both SpecDecode and Self-SpecDecode in both moderate size and large size models, achieving up to $1.53\times$ speedup compared to standard autoregressive decoding.

**SpecDecode (mostly) performs better when assisted with smaller models**. Table 1 shows that a smaller drafter model (*e.g.*, CodeLlama-7B-Instruct) generally leads to higher speedup for SpecDecode compared to a moderate-size model (*e.g.*, CodeLlama-13B-Instruct). This is mainly because smaller drafters have fewer parameters, requiring less computation and inference time to generate draft tokens, which are then passed to a large-scale verifier (*e.g.*, CodeLlama-34B) for parallel verification and correction, significantly reducing the overall time. One notable exception is in the text summarization task, where using the smaller model as the drafter yields a lower speedup than the moderate counterpart. We speculate that this is because smaller models like CodeLlama-7B-Instruct have limited capacity to handle extremely long texts, thus producing low-quality draft tokens, which can lead to a higher rejection ratio during verification, thereby increasing the overall latency.

Table 2: Consistency ratio comparison. In principle, all compared methods in our work are expected to guarantee output parity with standard autoregressive decoding as a result of the verification steps. However, due to numerical precision inaccuracies and potentially tied probabilities during the computation, the generation results might vary in practice and are subject to experimental environment and hardware specifications. To empirically validate the output consistency of all methods with vanilla autoregressive decoding, we report the consistency ratio for all methods to ensure a fair comparison under the same experimental environment.

| Method | Text Summarization (XSum) | Code Generation (HumanEval) | Mathematical Reasoning (GSM8K) |
|---|---|---|---|
| SpecDecode (Leviathan et al., 2023) | | | |
| CodeLlama-34B$_{INST}$ | | | |
| w/ drafter CodeLlama-13B$_{INST}$ | 94.98% | 78.56% | 93.91% |
| Self-SpecDecode (Zhang et al., 2024a) | | | |
| Llama2-7B$_{INST}$ | 94.26% | 94.83% | 94.14% |
| CodeLlama-13B$_{INST}$ | 92.53% | 95.20% | 93.78% |
| CodeLlama-34B$_{INST}$ | 93.31% | 97.79% | 94.16% |
| ParaDecode (Ours) | | | |
| Llama3.1-8B$_{INST}$ | 95.36% | 97.24% | 95.35% |
| CodeLlama-34B$_{INST}$ | 93.27% | 98.02% | 94.65% |

**Self-SpecDecode tends to achieve higher speedups as the model size increases**. By skipping a larger portion of intermediate layers, Self-SpecDecode can generate draft tokens much faster than the full model, thereby achieving significant speedup. However, this approach provides limited speedup improvement for mid-sized models such as Llama2-7B-Instruct. The reason is that such models have only 32 layers, skipping too many layers negatively impacts the quality of the draft tokens, which will lead to a high rejection rate during verification, and consequently increase the decoding latency. On the contrary, skipping too few layers does not yield sufficient speedup, as the performance gains stem primarily from reducing the computations associated with skipped layers.

**ParaDecode guarantees output parity with standard autoregressive decoding.** As shown in Table 2, we empirically evaluate the output parity guarantee of all baseline methods across three benchmarks. Despite theoretical guarantees, both SpecDecode and Self-SpecDecode fail to achieve 100% generation parity with standard autoregressive decoding. One possible reason is the numerical precision inaccuracies during inference, as we use BF16 precision for inference, which may cause the model to select different tokens compared to standard decoding.

## 4.2 ABLATION STUDY

Table 3: Ablation study on ParaDecode. We report the performance of our method by ablating the training of the lightweight LM head and analyzing the impact of the verification step. The table shows the resulting speedup and consistency ratio on the code generation task (HumanEval).

| Method | Consistency Ratio (%) | Speedup ($\times$) |
|---|---|---|
| ParaDecode | 97.24% | 1.312$\times$ |
| w/o verification step | 23.83% | 1.605$\times$ |
| w/o lightweight LM head training | 71.28% | 0.863$\times$ |

**The verification step is essential for ensuring output parity.** As shown in Table 3, As shown in Table 3, removing the verification step from ParaDecode significantly reduces the consistency ratio, even though it achieves a slightly higher speedup. Without verification, there is no control over the generated tokens, which can lead to ParaDecode producing entirely different content as early predictions may not always be reliable. This results in the acceptance of potentially incorrect early predicted tokens and causes significant deviations from standard autoregressive decoding. This finding underscores the importance of the verification step in our method.

**Simply applying the final-layer LM head for early prediction will not lead to speedup.** To validate the utility of our lightweight LM heads, we replaced the trained lightweight LM head with the

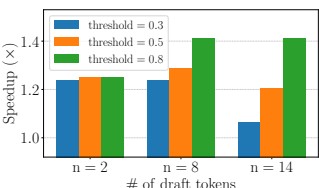
(a) SpecDec with 7B drafter.

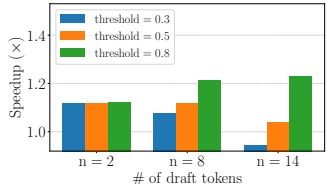
(b) SpecDec with 13B drafter.

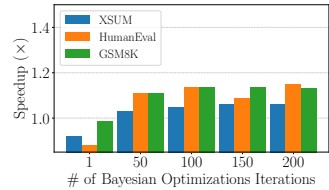
(c) Self-SpecDecode BO search.

Figure 5: Hyperparameter study of baseline models with CodeLlama-34B as the backbone model. (a) SpecDecode with CodeLlama-7B as drafter. (b) SpecDecode with CodeLlama-13B as drafter. (c) Self-SpecDecode optimized by different iterations of Bayesian optimization (BO) on three tasks.

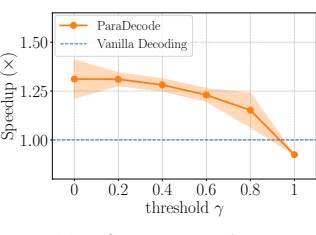
(a) Inference speedup.

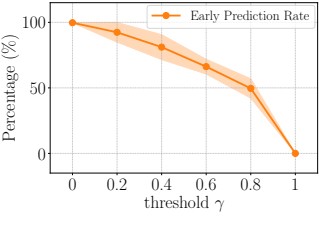
(b) Early prediction rate.

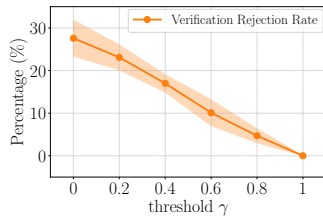
(c) Verification rejection rate

Figure 6: Hyperparamter study of ParaDecode by varying the early prediction threshold $\gamma$. This figure presents the evaluation results on HumanEval where (a) shows the speedup curve, (b) shows the early prediction rate, and (c) shows the verification rejection rate.

final-layer LM head for early prediction. As shown in Table 3, this ablation substantially compromises the consistency ratio and, even worse, results in a negative speedup. This is because applying the final-layer LM head to intermediate layers for early prediction only produces very low confidence (as illustrated in Figure 3a), which requires frequent corrections of the early predicted tokens as few early predictions are accepted—the resultant overheads significantly slow down the decoding process because once a token is rejected, the KV cache generated during parallel computation for the next token immediately becomes invalid, as they are based on an incorrect preceding token.

## 4.3 ANALYSIS

**SpecDecode is sensitive to hyper-parameter configurations.** Figure 5 presents the hyperparameter study for SpecDecode, indicating that it is challenging to achieve consistent speedup without careful tuning. In this study, we use CodeLlama-34B-Instruct as the backbone model, and performed a comprehensive hyperparameter search on the HumanEval benchmark, with CodeLlama-7B-Instruct and CodeLlama-13B-Instruct as the assistant models, respectively. We explored various configurations with different max number of draft tokens $n = [2, 8, 14]$ and confidence threshold $= [0.3, 0.5, 0.8]$. Figure 5a and Figure 5b reveal that significant speedup variance exists across different settings. Moreover, the variance becomes even larger when the confidence threshold shifts. Notably, when $n = 14$ and the threshold is 0.3, the speedup of 13B drafter is merely 0.944, which is even worse than standard decoding. These findings highlight that careful tuning hyperparameters is essential to optimize SpecDec's speedup performance and avoid potential slowdowns.

**Self-SpecDecode requires task-specific architecture for optimized speedup performance**. Self-SpecDecode has been proposed as a training-free, inference schema that could be employed on LMs in a plug-and-play manner. However, it requires an additional Bayesian optimization (BO) process to first obtain a set of layers to skip (as the drafter) before it can be adopted for inference. The time-consuming BO process greatly limits its practicality, moreover, it requires a set of examples as validation data to select the desired drafter. To further investigate the effect of this search process, we optimize CodeLlama-34B-Instruct with different numbers of BO iterations, as shown in Figure 5c. In general, more number of iterations could lead to improved evaluation performance. Yet, on HumanEval, it demonstrates a decrease in performance in the process. This implies scaling the number of iterations does not monotonically reflect better results, and the iteration process might require careful tuning to select an optimized drafter for inference.

**ParaDecode can work well without searching thresholds for early predictions.** As introduced in Equation (1), the hyperparameter $\gamma$ controls early predictions and triggers parallel token processing. To study its impact, we conduct experiments with Llama3.1-8B-Instruct, and Figure 6 presents the speedup results, early prediction rate, and verification rejection rate, with varying values of $\gamma = [0, 0.2, 0.4, 0.6, 0.8, 1]$. Specifically, $\gamma = 1$ effectively means no early predictions, as very few tokens have a probability greater than 1, while $\gamma = 0$ triggers early predictions at every inference step. It can be observed that the early prediction rate decreases with increasing $\gamma$, while the verification rejection rate also decreases, and the inference speed is jointly affected by both factors. Surprisingly, we find that $\gamma = 0$ leads to the fastest overall speedup, demonstrating that our method does not require hyperparameter tuning. Based on this finding, we set $\gamma = 0$ for all experiments without further tuning, and the results in Table 1 and Table 2 confirm the effectiveness of this setting. We attribute this to the fine-tuned lightweight LM head, which enables early predictions with high confidence, resulting in a lower verification rejection rate (around 30%, as presented in Figure 6c). This finding suggests that encouraging more early predictions is beneficial for speedup and demonstrates that our method can work efficiently and accurately without hyperparameter tuning, simply by triggering early predictions at every step with the fine-tuned lightweight LM head.

## 5 RELATED WORK

### 5.1 EARLY EXITING

Early exiting enables language models (LMs) to complete prediction at intermediate layers, reducing computational overhead and accelerating generation. Previous approaches achieved this by adding decision branches or language modeling heads at various depths (Teerapittayanon et al., 2016; Huang et al., 2018a; Elbayad et al., 2020; Schuster et al., 2022). Perhaps the most similar work to ours is (Yang et al., 2024), where the authors utilize multi-processing for pipelined decoding via early exiting. While conceptually similar, their approach diverges from ours in both method design and the implementation of parallelism—they need to train the last-layer LM head for early exiting, potentially deviating from the standard autoregressive decoding. Moreover, multi-processing incurs initialization overhead and communication costs, which also limits its practical efficiency. Another notable recent work is mixture-of-depth (Raposo et al., 2024), which dynamically skips transformer blocks to enhance efficiency, however it still lacks the output parity guarantee with the standard autoregressive decoding. For a more detailed discussion on this line of research, we refer the readers to Khoshnoodi et al. (2024).

### 5.2 SPECULATIVE DECODING

Speculative decoding (Leviathan et al., 2023; Chen et al., 2023) has emerged as an effective approach for speeding up language model generation. It aims to reduce decoding latency by drafting tokens using smaller auxiliary models and verifying multiple tokens in parallel with larger models, which has been actively investigated recently (He et al., 2024; Fu et al., 2024; Liu et al., 2024a; Cai et al., 2024; Liu et al., 2024b; Li et al., 2024; Qin et al., 2024). Among them, self-speculative decoding (Zhang et al., 2024a) addresses the limitation of requiring an auxiliary smaller model as the drafter. LayerSkip (Elhoushi et al., 2024) employs early exiting to generate draft tokens while continuing with the remaining layers for verification, thereby accelerating the decoding process. Other notable works also investigate handling long text sequences (Bae et al., 2023; Kim et al., 2023; Hooper et al., 2023). A comprehensive overview in this area can be found in Xia et al. (2024a).

## 6 CONCLUSION

In this work, we introduced ParaDecode, a new approach designed to accelerate autoregressive LM decoding while preserving output consistency. ParaDecode achieves this by generating the next tokens at intermediate layers using our newly introduced lightweight LM heads. These early token predictions trigger parallel token processing for the subsequent token position, utilizing batched matrix operations. A verification step ensures consistency with standard autoregressive outputs. Notably, ParaDecode demonstrates consistent improvements in decoding throughput across various generation tasks compared to baselines, while imposing minimal constraints (*e.g.*, no auxiliary model or fine-tuning of existing model parameters required). Further discussion on limitation and future works is presented in Appendix D.

## REPRODUCIBILITY STATEMENT

We strive to ensure the highest reproducibility of our work by providing (a) our code in the supplementary materials, and (b) the dataset, benchmark, and implementation details in Appendices B and C.

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

## A    PROOFS

**Lemma A.1.** *For any $\boldsymbol{E}^{(i)} \in \mathbb{R}^{|\mathcal{V}| \times d}$, there exists a $\boldsymbol{T}^{(i)} \in \mathbb{R}^{d \times d}$ such that $\boldsymbol{E}^{(i)} = \boldsymbol{E}^* \boldsymbol{T}^{(i)}$.*

*Proof.* We first prove that $\boldsymbol{E}^*$ (the last-layer LM head weights) is full-rank, and then present an explicit form of $\boldsymbol{T}^{(i)}$ that satisfies $\boldsymbol{E}^{(i)} = \boldsymbol{E}^* \boldsymbol{T}^{(i)}$. During LLM pretaining, the last-layer LM head weights $\boldsymbol{E}^*$ are trained together with last layer hidden states $\boldsymbol{H}^* \in \mathbb{R}^{|\mathcal{D}| \times d}$ ($|\mathcal{D}|$ is the total number of training tokens in the corpus) to learn the ground-truth next-token prediction probability $\boldsymbol{P}^* \in \mathbb{R}^{|\mathcal{D}| \times |\mathcal{V}|}$:
$$\boldsymbol{P}^* \approx \mathrm{Softmax}(\boldsymbol{H}^* \boldsymbol{E}^{*\top}).$$
As the softmax function cannot increase matrix rank (Kanai et al., 2018), we have $\mathrm{rank}(\mathrm{Softmax}(\boldsymbol{H}^* \boldsymbol{E}^{*\top})) \leq \mathrm{rank}(\boldsymbol{H}^* \boldsymbol{E}^{*\top}) \leq \min\{\mathrm{rank}(\boldsymbol{E}^*), \mathrm{rank}(\boldsymbol{H}^*)\} \leq \mathrm{rank}(\boldsymbol{E}^*)$. Thus, to achieve a good approximation of $\boldsymbol{P}^*$, $\mathrm{rank}(\boldsymbol{E}^*)$ must closely match $\mathrm{rank}(\boldsymbol{P}^*)$. Given the complexity and diversity of natural language, the empirical distribution $\boldsymbol{P}^*$ derived from the pretraining data is extremely high-rank (Yang et al., 2018). Therefore, to accurately model this high-rank distribution, $\boldsymbol{E}^*$ must be full-rank.

Given $\boldsymbol{E}^*$ being full-rank, $\boldsymbol{U} := \boldsymbol{E}^{*\top} \boldsymbol{E}^*$ is invertible. We can define $\boldsymbol{T}^{(i)}$ as $\boldsymbol{T}^{(i)} = \boldsymbol{U}^{-1} \boldsymbol{E}^{*\top} \boldsymbol{E}^{(i)}$, then:

$$\boldsymbol{E}^* \boldsymbol{T}^{(i)} = \boldsymbol{E}^* \boldsymbol{U}^{-1} \boldsymbol{E}^{*\top} \boldsymbol{E}^{(i)} = \left( \underbrace{\boldsymbol{E}^* (\boldsymbol{E}^{*\top} \boldsymbol{E}^*)^{-1} \boldsymbol{E}^{*\top}}_{=\boldsymbol{P}} \right) \boldsymbol{E}^{(i)}$$

Note that $\boldsymbol{P} = \boldsymbol{E}^* (\boldsymbol{E}^{*\top} \boldsymbol{E}^*)^{-1} \boldsymbol{E}^{*\top}$ is the projection matrix onto the column space of $\boldsymbol{E}^*$ (as $\boldsymbol{P}^2 = \boldsymbol{P}$). Since $\boldsymbol{E}^*$ is full rank, $\boldsymbol{E}^{(i)}$ lies in the column space of $\boldsymbol{E}^*$. Therefore, applying $\boldsymbol{P}$ to $\boldsymbol{E}^{(i)}$ gives $\boldsymbol{E}^{(i)}$ itself, confirming that $\boldsymbol{E}^{(i)}$ can be expressed as $\boldsymbol{E}^* \boldsymbol{T}^{(i)}$.

$\square$

## B    BENCHMARK DETAILS

We evaluate our method on a diverse set of text generation tasks, including text summarization, code generation, and mathematical reasoning, covering a broad spectrum of language model capabilities.

**Text summarization.** For text summarization, we use the widely adopted extreme summarization (XSum) dataset (Narayan et al., 2018), where the models are prompted to produce a single-sentence summary of a news article, testing their ability to identify and precisely summarize the most salient information in a coherent sentence. Following previous works (Zhang et al., 2024a), we randomly sample 1K instances from the test split for evaluation, and 10K instances from the training split for training the lightweight LM head.

**Code generation.** For code generation, we evaluate our method on the HumanEval (Chen et al., 2021) benchmark, which assesses Python programming skills through a variety of coding problems, ranging from basic tasks to complex problem-solving challenges. Since the standard HumanEval benchmark does not provide a training set, we use the entire MBPP (Austin et al., 2021) dataset for training, which contains a set of crowd-sourced Python programming problems designed to be solvable by entry-level programmers, covering programming fundamentals and standard library functionality. This results in a total of 974 training samples and 164 test samples for this task.

**Mathmatical reasoning.** We use GSM8K (Cobbe et al., 2021) as the benchmark for mathematical reasoning, which contains diverse grade-school math word problems created by human problem writers. The dataset consists of 7.5K training problems and 1K test problems. These problems typically require multiple reasoning steps to solve and involve performing a sequence of basic arithmetic operations (such as addition and subtraction) to arrive at the final answer. The goal of this task is specifically to evaluate the LLM's ability in multi-step mathematical reasoning.

## C  IMPLEMENTATION DETAILS

**Training details.** The lightweight LM heads in our method are trained through full-parameter fine-tuning using the alignment-handbook repository[2] with Nvidia H100 GPUs. Specifically, we utilize DeepSpeed ZeRO-3 (Rajbhandari et al., 2020) along with FlashAttention (Dao, 2024) for distributed training, and we enable BF16 mixed precision training to enhance training efficiency. We generate on-policy data to train the lightweight LM heads by prompting the off-the-shelf Llama3.1-8B-Instruct or CodeLlama-34B-Instruct to produce responses using greedy decoding for prompts from the mixed training split obtained from the benchmarks. By default, our models are trained using the Adam optimizer (Kingma & Ba, 2014) for 50 epochs, with a batch size of 128, a learning rate of 5e-3, and a cosine learning rate schedule with 3% warmup steps.

**Inference details.** During inference, we adopt the zero-shot evaluation by directly prompting the model to generate responses and apply the corresponding chat templates to format the prompts, as all backbone models used in our work are instruction-tuned versions. The framework is implemented using the HuggingFace Transformers library[3], and we utilize the standard greedy decoding strategy as the baseline for a reproducible comparison. Following Zhang et al. (2024a), the maximum number of new tokens is set to 512.

## D  LIMITATIONS AND FUTURE WORK

**Limitations.** Our work focuses on accelerating LM decoding without directly improving the quality of the outputs, so ParaDecode may encounter similar issues as general LM decoding, such as hallucinations (Huang et al., 2023) and generating unsafe content (Zhang et al., 2024b). Additionally, while ParaDecode consistently accelerates standard autoregressive decoding, it may occasionally incur higher FLOPs, particularly when an early-predicted token does not match the gold token, necessitating a reversion. However, such cases are rare in practice, and we find that strict parity with standard autoregressive decoding is not always necessary. The early-predicted tokens, even when they differ from the final prediction, are typically still meaningful. Therefore, one might consider relaxing strict consistency requirements to avoid unnecessary computational waste.

**Future work.** In the future, we plan to integrate ParaDecode with other efficiency-enhancing techniques like model pruning and quantization. Model pruning (Xia et al., 2024b) offer the potential to reduce model size and parameter space by identifying and removing less critical weights or neurons. When combined with ParaDecode, pruning could lead to even faster decoding times and lower memory usage without significantly impacting performance. Similarly, ParaDecode can also be seamlessly integrated with quantization (Dettmers et al., 2023), which reduces the precision of model weights and activations, substantially lowering memory and compute requirements. These directions could be particularly valuable for efficient inference on mobile and edge devices.

---

[2]https://github.com/huggingface/alignment-handbook
[3]https://github.com/huggingface/transformers

