# OpenReview forum: "Fast and Accurate Language Model Decoding via Parallel Token Processing"
_ICLR.cc/2025/Conference — Submitted to ICLR 2025_

### Official Review · Reviewer_vTw5 · 2024-11-02

**Soundness:** 1
**Presentation:** 3
**Contribution:** 1
**Rating:** 3
**Confidence:** 5

**Summary:**

This paper argues that in language model, some easy tokens can already be predicted using intermediate layer representations. Based on this observation, this paper proposes a novel decoding algorithm to accelerate LLM inference. It borrows the idea from speculative decoding by using output prediction from intermediate layer representation as draft tokens. Once a draft token is generated, the language model can immediate start predict the next token based on the draft token. Then it usese a verification scheme to ensure the accuracy of the generated tokens.

**Strengths:**

1. The paper is easy to understand and the figures are illustrative.

2. The observation that "some easy tokens can already be predicted using intermediate layer representations" is interesting.

**Weaknesses:**

1. Although the paper claims that the proposed method can accelerate LLM inference with an auxiliary model, it still requires training additional LM head, which introduces extra parameters and memory overhead.  I do not find specific data in the paper about how many extra parameters are introduced for the models used in the experiments.  But according to Section 2.1, each LM head for a 8B model introduces 16M parameters. And according to some previous studies [1,2]. Llama-2-7B and Llama-2-13B can be accelerated by a Llama-68M model, which is about 4 extra LM heads. So it seems that the proposed method does not have advantages in reducing memory overhead introduced by a small model.

2. I tried to analyze the theoretical upper bound of speed-up of the proposed method, and it does not look promising. Assume the tokens can be accurately predicted with first $\alpha$ (0<$\alpha$<1) of whole layers.  And assume we want to generate $N$ tokens in total. Let the latency of generate one token be $t$. Then based on the ideal case depicted in Figure 2, the total time to generate $N$ tokens is $((N-1)\alpha+1)t/N$, when $N$ approaches infinity, the speed up is $1/\alpha$. According to Figure 3(b) in the paper, I think $\alpha$ is about 0.5-0.75, so the proposed method can at best be 2 times faster than autoregressive decoding. And some speculative decoding with powerful drafters (e.g., Medusa and EAGLE) can achieve 4 times speed-up, while only introducing 370M extra parameters.

3. Even if $\alpha$ can be really small, the proposed method still has a potential problem with memory. In order to achieve the ideal case, the proposed method needs to compute $1/\alpha$ tokens in parallel, but these tokens are executed at different layers (as illustrated in Figure 3). And based on the batched matrix operations, the proposed method needs to load parameters of all $1/\alpha$ layers simultaneously, which introduces additional memory overhead compared to running a single layer. Notice that the speculative decoding does not have this problem. In a word, the batched matrix operation introduces additional overhead during computation, which might not be parallelized perfectly.

[1] Specinfer: Accelerating large language model serving with tree-based speculative inference and verification
[2] Multi-Token Joint Speculative Decoding for Accelerating Large Language Model Inference

**Questions:**

1. Can authors report how much additional parameters are introduced for each model used in the experiments?

2. In Table 1, why self-specdecode and the proposed method have different target models? They are not comparable in my opinion.

3. The early-prediction verification mechanism does not make sense to me. i think it only works for deterministic greedy setting. In stochastic setting, even when the two distributions are same, there is still a high chance that t and t* are different. For example, p_i=(0,5,0.5) and p*=(0.5,0.5), then there is a 50% chance of rejection.

4. I think the layout of Table 1 needs significant improvement.

5. I think the choices of draft models for SpecDecode in the experiments are bad. The draft model needs to be significant faster than the target model, but 7B and 13B model are not significant faster than 34B model, so naturally SpecDecode cannot be faster. I suggest authors following previous studies (e.g., [1,2]) to choose appropriate target and draft model (e.g., Llama-68M and Llama-2-7B, Llama-2-7B and Llama-2-70B, Llama-68M and Llama-2-13B).

---

> ### Author Response · Authors · 2024-11-29
> **Response to Reviewer vTw5 [part 1/3]**
>
> We sincerely appreciate the reviewer's insightful feedback, which has significantly enhanced the quality of our work. We have also added discussions with the suggested papers in our updated manuscript.
>
>
> **[Q1]**: Although the paper claims that the proposed method can accelerate LLM inference with an auxiliary model, it still requires training additional LM head, which introduces extra parameters and memory overhead. I do not find specific data in the paper about how many extra parameters are introduced for the models used in the experiments. But according to Section 2.1, each LM head for a 8B model introduces 16M parameters. And according to some previous studies [1,2]. Llama-2-7B and Llama-2-13B can be accelerated by a Llama-68M model, which is about 4 extra LM heads. So it seems that the proposed method does not have advantages in reducing memory overhead introduced by a small model.
>
> **[A1]**: Compared to using an additional draft model, the advantages of our ParaDecode can be summarized below:
>
>
>
> - **Easily adjustable memory overhead**: The additional overhead introduced by our lightweight LM head is actually quite minimal and flexible – it can be adjusted by varying the number of intermediate heads.  In our implementation, we only use a single lightweight LM head (with 16M parameters) positioned at the middle layer (e.g., 16-th layer for a model with 32 layers) of the model, which is sufficient enough to confidently make reliable early predictions, as illustrated in **Figure 3(b)**. Therefore, our memory overhead (16M) is significantly smaller than introducing an additional draft model (68M), making our approach *parameter-efficient*.
>
> - **Efficient training**: Additionally, our method is also *training-efficient*. For instance, pre-training a draft model from scratch requires 275 GPU hours [1], whereas our lightweight LM head can be fine-tuned in just about 2 GPU hours.
>
> **[Q2]**: According to Figure 3(b) in the paper, I think $\alpha$ is about 0.5-0.75, so the proposed method can at best be 2 times faster than autoregressive decoding. And some speculative decoding with powerful drafters (e.g., Medusa and EAGLE) can achieve 4 times speed-up, while only introducing 370M extra parameters.
>
> **[A2]**: We appreciate the reviewer’s thoughtful analysis and would like to clarify that Figure 3(b) is just an illustration of the early predictions with the 8B model, and the value of $\alpha$ can **vary with model size**. For instance, for a larger model with 48 layers (e.g., CodeLlama-34B), the 12th layer is good enough to make promising early predictions, and the corresponding $\alpha$ can be as low as 0.25, which leads to higher speed-up potential.
>
> While the previous work such as Medusa (specifically, Medusa-1) is reported to achieve a speedup of 2.1x without compromising generation quality in some cases [3], its practical performance can vary and may be as low as 1.29x depending on the application scenarios [4]. Additionally, Medusa-2 does not guarantee output parity with standard autoregressive decoding, making it difficult to compare directly with our method, which ensures output consistency with autoregressive decoding.
>
> In contrast,  our method introduces only 16M parameters, which is significantly more efficient than previous works that require much larger draft models [5] or additional LM heads [3], while still achieving considerable speedups (e.g., 1.42x speedup for GSM8K).
>
>
> **[Q3]**: Even if $\alpha$ can be really small, the proposed method still has a potential problem with memory. In order to achieve the ideal case, the proposed method needs to compute
> $1 / \alpha$ tokens in parallel, but these tokens are executed at different layers (as illustrated in Figure 3). And based on the batched matrix operations, the proposed method needs to load parameters of all $1/ \alpha$ layers simultaneously, which introduces additional memory overhead compared to running a single layer. Notice that the speculative decoding does not have this problem. In a word, the batched matrix operation introduces additional overhead during computation, which might not be parallelized perfectly.
>
> **[A3]**: We'd like to clarify that the entire model is already loaded into memory before the decoding process starts, as is common in standard inference procedures [1, 2, 3], and there is no need to repeatedly load parameters for each individual layer during the decoding process.
>
> In other words, since the model’s full set of parameters is already available in memory,  our method can simply fetch the corresponding weights from memory for parallel computation without any additional memory overheads compared to the standard decoding method.

---

> ### Author Response · Authors · 2024-11-29
> **Response to Reviewer vTw5 [part 2/3]**
>
> **[Q4]**: Can authors report how much additional parameters are introduced for each model used in the experiments?
>
> **[A4]**: The additional parameters introduced are merely $d \times d$, where $d$ is the hidden size. Specifically, for an 8B model with a hidden size of 4096, the additional parameters amount to approximately 16M. Similarly, for a 34B model with a hidden size of 8192, the additional parameters are around 64M.
>
> This is because we only use a single lightweight LM head at the middle layer of the model (e.g., the 16th layer for a model with 32 layers) in our implementation. This design choice is based on our observation that intermediate LM heads inserted at very shallow layers (e.g., layer 1 or 2) struggle to provide reasonable token predictions due to insufficient contextual information, while those inserted at very deep layers offer minimal speedup benefits as they are too close to the final layer. Therefore, we opt to insert only one additional LM head at the middle layer to balance prediction accuracy and computational efficiency.
>
> On the other hand, our method is flexible and can be adjusted for different models. For instance, models with more layers could benefit from additional intermediate heads to further enhance performance while maintaining efficiency (e.g., placing heads at the 20/40/60th layers for an 80-layer model). This adaptability allows our method to be tailored to the specific architecture and requirements of various models.
>
> **[Q5]**: In Table 1, why self-specdecode and the proposed method have different target models? They are not comparable in my opinion.
>
> **[A5]**: We appreciate the reviewer’s comments. The reason is that Self-SpecDecode [6] relies on model-specific configurations (e.g., the skipped intermediate layers for each model are pre-configured before the decoding process), and its open-source implementation was highly intertwined with obsolete versions of libraries that only supported CodeLlama and Llama 2 (e.g., the authors recommended using transformers version v4.33.1 to ensure compatibility with their codebase in their GitHub [issue-8](https://github.com/dilab-zju/self-speculative-decoding/issues/8#issuecomment-1860190005) and [issue-14](https://github.com/dilab-zju/self-speculative-decoding/issues/14#issuecomment-1996249908), while the support for Llama 3.1 was not introduced until transformers version v4.43.0).
> These practical limitations restrict the direct applicability of their codebase to newer models due to:
> - (1) The lack of implementation of essential functionalities (e.g., grouped-query attention, RoPE scaling), which necessitates substantial development to the codebase to support Llama 3.1.
> - (2) The requirement of an extensive Bayesian Optimization search to determine the configurations for specific models, as suggested in this [GitHub issue](https://github.com/dilab-zju/self-speculative-decoding/issues/20).
>
> Given the above challenges in establishing a fair comparison, we opted to compare ParaDecode and Self-SpecDecode using the same CodeLlama-34B as the target model. The results in **Table 1** demonstrate that our method consistently achieves superior speedups across three benchmarks, highlighting its effectiveness.

---

> ### Author Response · Authors · 2024-11-29
> **Response to Reviewer vTw5 [part 3/3]**
>
> **[Q6]**: The early-prediction verification mechanism does not make sense to me. i think it only works for deterministic greedy setting. In stochastic setting, even when the two distributions are same, there is still a high chance that t and t* are different. For example, p_i=(0.5,0.5) and p*=(0.5,0.5), then there is a 50% chance of rejection.
>
> **[A6]**: We acknowledge that our current implementation aligns more naturally with the deterministic greedy sampling setting. However, we would like to point out that our approach can also be extended to the stochastic setting by employing rejection sampling [8,9] as the verification strategy. This strategy has been theoretically proven to ensure parity with standard autoregressive decoding (please see Appendix A of [8]), even when using non-deterministic sampling methods such as temperature-based sampling.
>
> While our current verification mechanism (without the rejection sampling strategy) might occasionally reject correct tokens (i.e., false negative), it will never accept incorrect ones (i.e., false positive). This ensures the generation quality, despite potential extra latency for correcting false negatives. However, we would like to note that:
> - **False negatives are quite rare**: The reason is that well-aligned LLMs tend to make high-confidence token predictions [10], making the token distribution quite concentrated (e.g., the max probability is typically close to 1.0 as shown in **Figure 3(b)**). This narrow distribution suggests that most of the probability is allocated to a single token.
> - **Stochastic sampling probably yields the same output**: Following the reviewer's example, the cases like $p_i = p^* = (0.5, 0.5)$ will occur very rarely in practice based on the above analysis. Instead, it is more likely that $p_i \approx p^* = (0.9, 0.1)$, making it highly probable that sampling from $p_i$ and $p^*$ will yield the same outcome.
>
> Despite the conceptual advantage of rejection sampling, the additional latency associated with it may offset the speedup by reducing false negatives in practice. Therefore, our current simple verification strategy can achieve comparable or even better speedups compared to rejection sampling-based verification, while being flexible to seamlessly integrate rejection sampling as described in [8] if needed.
>
> **[Q7]**: I think the choices of draft models for SpecDecode in the experiments are bad. The draft model needs to be significant faster than the target model, but 7B and 13B model are not significant faster than 34B model, so naturally SpecDecode cannot be faster. I suggest authors following previous studies (e.g., [1,2]) to choose appropriate target and draft model (e.g., Llama-68M and Llama-2-7B, Llama-2-7B and Llama-2-70B, Llama-68M and Llama-2-13B).
>
> **[A7]**: Thank you for the suggestion, we’ve added discussions on these works in Section 5.2. However, we would like to clarify that we directly follow the setups of previous works [4,5] to decide our choice of drafter/verifier models.
>
> We agree that using a separately trained smaller draft model might potentially lead to higher speedups for SpecDecode. However, these small draft models are usually tailored for specific target models and lack compatibility with newer models. For instance, Llama-68M is compatible only with the Llama-2 series and not with newer models like Llama-3, due to differences in vocabulary and tokenizer configurations.
>
> Furthermore, it is also impractical to assume the availability of such "68M" models for every newly released LLM – these "suitable" small drafter models are typically unavailable in practice, and training a model from scratch requires massive computational resources (as stated in our response to Q1). In other words, using such an ad-hoc LLaMA-68M as the drafter is an idealized setup and may yield overly optimistic speedup estimations for speculative decoding, which is unlikely to be achievable in practice given the rapid release of LLMs nowadays.
>
>
> Please let us know if you have any further questions, and we are happy to incorporate additional suggestions you might have! If you find our response satisfactory, we would be grateful if you could consider raising your score. Thanks again for your time and efforts!

---

> ### Author Response · Authors · 2024-11-29
> **References**
>
> **References**
> - [1] Miao et al. SpecInfer: Accelerating Generative Large Language Model Serving with Tree-based Speculative Inference and Verification. ASPLOS 2024.
> - [2] Qin et al. Multi-Token Joint Speculative Decoding for Accelerating Large Language Model Inference. arXiv:2407.09722
> - [3] Cai et al. Medusa: Simple Framework for Accelerating LLM Generation with Multiple Decoding Heads. ICML 2024.
> - [4] Xia et al. Spec-Bench: A Comprehensive Benchmark and Unified Evaluation Platform for Speculative Decoding. ACL 2024.
> - [5] Li et al. EAGLE: Speculative Sampling Requires Rethinking Feature Uncertainty. ICML 2024.
> - [6] Zhang et al. Draft & Verify: Lossless Large Language Model Acceleration via Self-Speculative Decoding. ACL 2024.
> - [7] Elhoushi et al. LayerSkip: Enabling Early Exit Inference and Self-Speculative Decoding. ACL 2024.
> - [8] Leviathan et al. Fast Inference from Transformers via Speculative Decoding. ICML 2023.
> - [9] Chen et al. Accelerating Large Language Model Decoding with Speculative Sampling. arXiv:2302.01318
> - [10] Tian et al. Just Ask for Calibration: Strategies for Eliciting Calibrated Confidence Scores from Language Models Fine-Tuned with Human Feedback. EMNLP 2023.

---

> > ### Comment · Reviewer_vTw5 · 2024-11-30
> >
> > Thank you for providing the detailed rebuttal and clarifications. However, I still have some remaining concerns and questions that I would like to raise for further clarification:
> >
> > # Q1:
> >  If there is only one additional head, the number of new parameters is indeed smaller, which is noted. However, this introduces two new questions:
> > 1. There seems to be a mismatch between the method described in the paper and its implementation. Based on Figure 1 and the description in Section 2, my understanding is that more than one additional head is utilized. In particular, the figure clearly suggest that multiple layers have additional heads. The current experimental results do not evaluate the method’s performance when multiple heads are used, which is inadequate. If the authors intend to focus solely on the case with a single additional head, the writing has to be revised accordingly for consistency.
> > 2. EAGLE introduces 560M new parameters to a 34B model, while the proposed method introduces 64M. While the number of new parameters is reduced, considering the original model size (34B) the overall inference cost—such as memory usage and computational time—is unlikely to be significantly different between EAGLE and the proposed method. As such, I am not fully convinced that reducing the number of new parameters alone is a strong enough motivation.
> >
> > # Q2:
> >
> >  Could the authors provide more details on how many tokens are generated by the intermediate head and accepted?
> >
> > # Q3:
> >
> >  There appears to be some misunderstanding regarding “loading parameters.” I understand that model weights are loaded onto the GPU. However, similar to the hierarchical memory architecture of CPUs, GPU memory has different levels, including global off-chip memory (analogous to main memory in CPUs) and local on-chip memory (similar to CPU caches like L1). GPU compute units cannot directly process data stored in global memory. Instead, the data must be moved from global memory to local memory (like moving data from CPU main memory to cache), which is the main bottleneck of LLM inference [1]. In standard decoding and speculative decoding, GPU only needs to load 1 layer to on-chip memory. But in the proposed method, GPU needs to load more than 1 layers to on-chip memory simultaneously. So I believe the proposed method increases the overheard of such data transfers, potentially preventing perfect parallelization and affecting inference efficiency. Could the authors elaborate further on this?
> >
> > # Q6:
> >
> >  The authors’ argument that “false negatives are rare” remains unconvincing. It relies on the assumption that the target distribution is highly concentrated on one token, resulting in a probability close to 1. However, natural language often features multiple words with similar probabilities, especially when synonyms or alternative phrasings exist. For instance, one could say “achieve an average speed-up of 1.2x” or “achieve an average acceleration of 1.2x.” Both are equally valid. Therefore, I believe it is common for the target distribution to include multiple words with similar probabilities, and this needs to be addressed more thoroughly.
> > # Q7:
> >
> >  I disagree with the assertion that the paper’s setup fully aligns with the experimental settings of SpecInfer and EAGLE. While configurations like (7B, 34B) are included in their experiments and indeed show speculative decoding performing poorly, there are also setups where speculative decoding works well (e.g., 68M and 7B, 7B and 70B, 8B and 70B). These settings have not been considered in the comparisons. As mentioned in my original comment, the chosen setup seems to inherently disadvantage speculative decoding. Therefore, I am not fully convinced that the current experimental results provide a solid foundation for the claims made in the paper.
> >
> > [1] Fast Inference from Transformers via Speculative Decoding

---

> > > ### Author Response · Authors · 2024-12-04
> > > **Response to Reviewer vTw5 [Round 2, Part 1/3]**
> > >
> > > We would like to express our gratitude to reviewer vTw5 for taking the time to provide thoughtful feedback and a candid exchange of opinions about our work. We appreciate the reviewer’s insights and suggestions, which significantly helped us improve the presentation and positioning of our research.
> > >
> > > Although we realize that the discussion stage is closing soon and we may not be able to receive any further feedback from the reviewer, we'd like to take this final opportunity to address your comments and provide clarifications. Regardless of the final decision, we hope our responses demonstrate our appreciation for your insights and our commitment to improving our work. Thank you again for your efforts and for engaging with our submission.
> > >
> > > **[Round-2 Q1]**:  If there is only one additional head, the number of new parameters is indeed smaller, which is noted. However, this introduces two new questions:
> > >
> > > There seems to be a mismatch between the method described in the paper and its implementation. Based on Figure 1 and the description in Section 2, my understanding is that more than one additional head is utilized. In particular, the figure clearly suggest that multiple layers have additional heads. The current experimental results do not evaluate the method’s performance when multiple heads are used, which is inadequate. If the authors intend to focus solely on the case with a single additional head, the writing has to be revised accordingly for consistency.
> > >
> > >
> > > EAGLE introduces 560M new parameters to a 34B model, while the proposed method introduces 64M. While the number of new parameters is reduced, considering the original model size (34B) the overall inference cost—such as memory usage and computational time—is unlikely to be significantly different between EAGLE and the proposed method. As such, I am not fully convinced that reducing the number of new parameters alone is a strong enough motivation.
> > >
> > > **[Round-2 A1]**: We appreciate the reviewers’ thoughtful comments. We would like to clarify that the proposed ParaDecode is a general framework that can accommodate a flexible number of lightweight LM heads. Figure 1 provides an illustration of ParaDecode in the context of using multiple lightweight LM heads to demonstrate its potential.
> > >
> > > In practice, we need to trade off training overheads and inference efficiency, as adding more LM heads introduces additional parameters. While we acknowledge that ParaDecode can achieve further performance boosts with multiple LM heads, our empirical findings indicate that a single LM head is sufficient to provide meaningful speed-ups. We will update the implementation details in the paper to enhance clarity.
> > > Furthermore, we would like to note that memory reduction is just a side benefit of our method – The primary motivation is to enhance the utilization of GPU parallelism, rather than merely reducing memory requirements. Compared to EAGLE, our method eliminates the reliance on a separate drafter model and introduces significantly fewer new parameters, making our approach both *parameter-efficient* and *training-efficient*: Specifically, we introduce 64M new parameters, whereas EAGLE introduces 560M. Additionally, EAGLE requires up to 4x48 GPU hours of training [10], while our method only needs 2 GPU hours.
> > >
> > > **[Round-2 Q2]**:  Could the authors provide more details on how many tokens are generated by the intermediate head and accepted?
> > >
> > > **[Round-2 A2]**: As indicated in Equation (1), the number of tokens generated by the intermediate LM head primarily depends on the threshold setting. For instance, if the threshold is set to 0, then all tokens are generated by the intermediate head. To comprehensively study its impact, please refer to Figure 6 (b) and (c), where we present the proportion of early predicted tokens and acceptance/rejection ratio with varying threshold values.

---

> > > ### Author Response · Authors · 2024-12-04
> > > **References [Round 2]**
> > >
> > > **References**
> > > - [1] Su et al. The Synergy of Speculative Decoding and Batching in Serving Large Language Models. arXiv:2310.18813
> > > - [2] Kim et al. SqueezeLLM: Dense-and-Sparse Quantization. ICML 2024.
> > > - [3] Huang et al. SwapAdvisor: Pushing Deep Learning Beyond the GPU Memory Limit via Smart Swapping. ASPLOS 2020.
> > > - [4] Jain et al. Checkmate: Breaking the Memory Wall with Optimal Tensor Rematerialization. MLSys 2020.
> > > - [5] Patil et al. POET: Training Neural Networks on Tiny Devices with Integrated Rematerialization and Paging. ICML 2022.
> > > - [6] Kwon et al. Efficient Memory Management for Large Language Model Serving with PagedAttention. SOSP 2023
> > > - [7] Dao et al. FlashAttention: Fast and Memory-Efficient Exact Attention with IO-Awareness. NeurIPS 2022
> > > - [8] Zhang et al. FlattenQuant: Breaking Through the Inference Compute-bound for Large Language Models with Per-tensor Quantization. COLING 2024.
> > > - [9] Wang et al. LightSeq: A High Performance Inference Library for Transformers. NAACL 2021.
> > > - [10]  Li et al. EAGLE: Speculative Sampling Requires Rethinking Feature Uncertainty. ICML 2024.
> > > - ​​[11] Leng et al. Taming Overconfidence in LLMs: Reward Calibration in RLHF. arXiv:2410.09724.
> > > - [12] Kadavath et al. Language Models (Mostly) Know What They Know. arXiv:2207.05221
> > > - [13] OpenAI. Gpt-4 technical report, 2023.
> > > - [14] Miao et al. SpecInfer: Accelerating Generative Large Language Model Serving with Tree-based Speculative Inference and Verification. ASPLOS 2024.
> > > - [15] Cai et al. Medusa: Simple Framework for Accelerating LLM Generation with Multiple Decoding Heads. ICML 2024.

---

> ### Author Response · Authors · 2024-12-04
> **Response to Reviewer vTw5 [Round 2, Part 2/3]**
>
> **[Round-2 Q3]**:  I believe the proposed method increases the overheard of such data transfers, potentially preventing perfect parallelization and affecting inference efficiency. Could the authors elaborate further on this?
>
> **[Round-2 A3]**: We acknowledge that batching and speculative decoding are two orthogonal approaches to improve GPU utilization in LLM inference [1]. To further justify the speedups achieved by our method, we would like to provide the following clarifications.
>
> ## Memory access latency can be reduced to a comparable level of compute latency
>
> - **Memory access latency is hardware-dependent and drastically reduced in the latest GPUs**: It is worth noting that early works [2] concluded that memory latency was much greater than computer latency on less performant GPUs such as A5000, which have limited memory bandwidth and smaller on-chip caches. Recent GPUs such as the H100/H200 have featured **increased memory bandwidth** (e.g., H100 is 5x faster than the A5000 in memory access), **larger on-chip caches** (e.g., H100’s L2 cache is 12.5x larger than the A5000’s), and **improved data transfer protocols** (e.g., H100 can prefetch data into cache while executing computations), all of which significantly reduce memory access latency. In our work, we conduct experiments on H100 GPUs with much higher bandwidth, making it possible for ParaDecode to achieve meaningful speedups by increasing computational throughput via the batching technique. This suggests the possibility that **memory access may no longer be the sole or primary bottleneck in LLM inference on high-end GPU architectures**.
>
> - **Memory access operations can be further optimized via software-level mechanisms**: In addition to hardware improvements, recent advances have introduced multiple techniques to further reduce memory overhead, such as swapping [3], recomputation [4], and their combination [5]. Notable works include vLLM [6], which introduces a novel memory management technique called PageAttention to optimize memory access operations. FlashAttention [7] applies tiling and kernel optimizations to reduce the peak memory required for attention computations and minimize memory I/O costs.
> ## Why can ParaDecode achieve meaningful speedups?
> While the above advances have significantly reduced memory access latency, we acknowledge that LLM inference can remain memory-bandwidth bound given its limited arithmetic intensity [2]. This allows ParaDecode to further increase computational throughput by parallelizing computation across multiple tokens via batching, maximizing the utilization of GPU tensor cores and reducing the compute latency. Moreover, the batched operations also save other overheads such as kernel launches and function calls, as sequential operations typically involve such repeated non-computational latency.
> Despite being memory-bandwidth bound, we believe the non-trivial speedups achieved by ParaDecode could potentially serve as an empirical evidence for the argument that **memory latency is significantly reduced to a level comparable to or even less impactful than compute latency on modern GPUs**, as suggested by the aforementioned GPU advancements. This phenomenon is likely to become more evident in the future, as we can expect compute latency in LLMs to further increase with models generating longer sequences. For example, recent state-of-the-art LLMs such as OpenAI’s o1, tend to produce long chains of thought for complex reasoning.
>
> Indeed, this implication is consistent with previous findings [8], which also observed that compute latency can be much larger than memory latency during LLM inference:
> > “However, as the input batch size and the sequence length increase, the compute-bound factor becomes predominant, overshadowing the influence of memory-bound. In such cases, matrix multiplication consumes up to 80% of the inference time, as reported by LightSeq [9].”
>
> ## Summary
> To summarize, our finding suggests that even LLM inference is theoretically memory-bound, in practice **reducing compute and memory latency could be both important** because of the complicated interplay between computation and memory access operations. For example, in the LLM forward passes, we need to frequently switch between memory fetching and matrix multiplications, leading to imperfect overlap of computation and memory access that jointly affects the overall inference efficiency. In other words, provided that arithmetic operations are potentially playing an increasingly critical role in overall inference efficiency on modern GPUs, it is entirely possible that optimizing compute latency can yield significant practical improvements in throughput. We will include more in-depth experiments to analyze the latency of memory access versus arithmetic operations in our next revision.

---

> ### Author Response · Authors · 2024-12-04
> **Response to Reviewer vTw5 [Round 2, Part 3/3]**
>
> **[Round-2 Q6]**:  The authors’ argument that “false negatives are rare” remains unconvincing. It relies on the assumption that the target distribution is highly concentrated on one token, resulting in a probability close to 1. However, natural language often features multiple words with similar probabilities, especially when synonyms or alternative phrasings exist. For instance, one could say “achieve an average speed-up of 1.2x” or “achieve an average acceleration of 1.2x.” Both are equally valid. Therefore, I believe it is common for the target distribution to include multiple words with similar probabilities, and this needs to be addressed more thoroughly.
>
> **[Round-2 A6]**: We would like to clarify that our argument regarding the concentrated output distribution of LLMs is not an assumption but an empirical observation. Specifically, we found that LLMs (particularly those fine-tuned with RLHF or instruction tuning) tend to be overconfident with concentrated output distributions. This finding is also observed and widely recognized in prior works. For example, a recent work [11] mentions that:
> > "Recent studies [12,13] show that RLHF-trained LLMs tend to exhibit overconfidence, potentially due to sharpened output distributions."
>
> We agree that rejection sampling can mitigate false negative cases. However, the impact of such cases on overall performance is trivial because they are rare in practice. Per the reviewer’s suggestion, we implemented a new variant of ParaDecode using rejection sampling as the verification strategy. The results on GSM8K show that this variant achieved similar speedups compared to our original approach with naive verification. Specifically, when using Llama3.1-8B-Instruct as the backbone model, our original method achieved a 1.42x speedup, while the rejection sampling variant achieved a 1.44x speedup.
>
>
> This result confirms our observation that false negatives have minimal impact on performance due to the concentrated nature of the output distribution.
>
>
> **[Round-2 Q7]**:  I disagree with the assertion that the paper’s setup fully aligns with the experimental settings of SpecInfer and EAGLE. While configurations like (7B, 34B) are included in their experiments and indeed show speculative decoding performing poorly, there are also setups where speculative decoding works well (e.g., 68M and 7B, 7B and 70B, 8B and 70B). These settings have not been considered in the comparisons. As mentioned in my original comment, the chosen setup seems to inherently disadvantage speculative decoding. Therefore, I am not fully convinced that the current experimental results provide a solid foundation for the claims made in the paper.
>
> **[Round-2 A7]**: We appreciate the reviewer’s feedback and have conducted additional baseline experiments as suggested. Following the experimental setup of SpecInfer [14], we evaluated speculative decoding using Llama2-68M as the drafter and Llama2-7B as the target model on the GSM8K task. The results show that speculative decoding achieves a 1.47x speedup, while our ParaDecode implemented with Llama3.1-8B achieves a 1.42x speedup on GSM8K, as indicated in Table 1. **However, this specific speedup result with Llama2-7B is not directly applicable to Llama3.1-8B**, as there is no equivalent Llama3.1-68M model available to pair with Llama3.1-8B, as previously discussed in our response to Q7.
>
> We acknowledge that speculative decoding may achieve higher speedups when using larger target models. However, employing an 8B speculator with a 70B target model would require loading both models into GPU memory simultaneously, which poses significant resource constraints. For example, this setup would require at least 156 GB of GPU memory (even with half-precision), exceeding the capacity of most high-end GPUs (e.g., A100/H100, which have 80 GB of memory).
>
> Furthermore, we would like to emphasize that speedup is not the sole benefit of our method. Other factors such as practicality and implementation complexity are also crucial for deploying accelerated LLM generation techniques in real-world applications. For instance, if an acceleration method requires several days or even weeks to set up (e.g., separately training a series of drafter model from scratch using the same tokenizer as the main model [14]), it becomes challenging to keep up with the rapid pace of advancements in new LLMs, which are being released at an increasingly frequent rate – currently on a near-weekly basis.
>
> Compared to previous works [10, 14, 15], our method achieves meaningful speedups through straightforward batching techniques while also offering flexible control over the additional parameters. To summarize, our method is not only easy to implement but also parameter-efficient and training-efficient (as detailed in our previous response to Q1), offering a lightweight yet practical solution for accelerating LLM inference.

---

### Official Review · Reviewer_XpyW · 2024-11-03

**Soundness:** 3
**Presentation:** 4
**Contribution:** 3
**Rating:** 5
**Confidence:** 3

**Summary:**

In this work, the authors propose to combine layer-skipping based decoding with parallel cache computation for the purpose of efficient and accurate text generation. Traditional layer-skipping decoding schemes do not ensure parity with standard auto-regressive decoding as they require modification to the model weights / computation performed. To perform ParaDecoding, light-weight LM heads are attached to intermediate model layers. If an intermediate LM head is sufficiently confident in a next token prediction, ParaDecode takes that token as the output and continues on to start decoding the next token. In order to preserve the KV cache, in parallel to decoding the next token in the sequence, they continue to compute the representation of the early-exited tokens. ParaDecode also employs a verification scheme where the early predicted token’s identity is verified against the token that would have been predicted had decoding not exited early.

**Strengths:**

* The ParaDecode methodology is interesting and novel, allowing arbitrary pretrained language models to benefit from layer-skipping decoding while maintaining consistency.

* The memory saving lightweight LM head methodology seems novel and broadly useful for other techniques that require additional LM heads such as Medusa decoding [1].

* The presentation of the ParaDecode method is very clear throughout the paper. The figures describing the method (Figure 1, Figure 4) make the method very explicit.

[1] Cai, Tianle, et al. "Medusa: Simple llm inference acceleration framework with multiple decoding heads." arXiv preprint arXiv:2401.10774 (2024).

**Weaknesses:**

* The vanilla SpecDecode baseline does not seem like a fair comparison on tasks other than code as the ParaDecode LM heads are trained on each task while a frozen CodeLlama model is used as the drafter for SpecDecode.

* There is no discussion or comparison with draft-head based speculative decoding (see Medusa as mentioned above) even though such methods achieve high speed ups while not requiring separate pre-trained draft models.

* While non-greedy sampling is an important setting for language generation, the paper only looks at greedy verification strategies. If possible it would be useful to include results for how ParaDecode performs with non-greedy sampling.

**Questions:**

* Would you be able to speak more towards how ParaDecode achieves a speedup when different layers still need to be fetched from memory for each token being decoded in parallel. While I understand that the actual computation is batched, as each token has a corresponding unique weight operating on it, it seems as though the weight movement per token generated would not be decreased. Any clarification would be appreciated.

* Tree based speculative decoding ([1], [2]]) sets the current state-of-the-art for speculative decoding techniques. Would tree based decoding work for ParaDecode?

[2] Miao, Xupeng, et al. "SpecInfer: Accelerating Generative Large Language Model Serving with Tree-based Speculative Inference and Verification." arXiv preprint arXiv:2305.09781 (2023).

---

> ### Author Response · Authors · 2024-11-29
> **Response to Reviewer XpyW [part 1/2]**
>
> We sincerely thank the reviewer for the constructive comments. We have updated the manuscript to incorporate the reviewer’s suggestions, which have greatly helped improve our work.
>
> **[Q1]**: The vanilla SpecDecode baseline does not seem like a fair comparison on tasks other than code as the ParaDecode LM heads are trained on each task while a frozen CodeLlama model is used as the drafter for SpecDecode.
>
> **[A1]**: As stated in **Appendix C**, the training of the lightweight LM head is based on responses generated by the model itself, rather than ground-truth responses from the benchmarks. This ensures that no new knowledge is introduced to the model. Furthermore, our lightweight LM head is not optimized for any specific task. Instead, it is trained on a mixture of on-policy data generated by the model itself using all task prompts (excluding their original responses). Therefore, we believe this provides a fair comparison with the vanilla SpecDecode baseline.
>
> **[Q2]**: There is no discussion or comparison with draft-head based speculative decoding (see Medusa as mentioned above) even though such methods achieve high speed ups while not requiring separate pre-trained draft models.
>
> **[A2]**: Thank you for the suggestion. We have included a discussion of Medusa in the updated manuscript (**Section 5.2**). Previous works like Medusa require training multiple full LM heads (e.g., 5 heads as suggested in the paper), each being a matrix of size $d \times vocab\\_{size}$. Specifically, for a Llama-3-8B-based Medusa instance with 5 heads, the additional parameters can be up to $5 \times 4096 \times 128,256$ = 2.63B parameters, which is approximately 156 times larger than the parameter count in our method – as stated in **Section 2.1 (Line 202)**, our approach only requires $4096 \times 4096$ = 16M parameters for the lightweight LM head, making it significantly more efficient.
>
>
> Moreover, training Medusa’s LM heads requires at least 5 GPU hours for the basic implementation (Medusa-1) [1], and the full Medusa model (Medusa-2) involves fine-tuning the entire model along with the Medusa heads trained in the first stage, resulting in even larger training overheads (e.g., the two-stage training can take 20 GPU hours). In contrast, training the lightweight LM head in our ParaDecode can be completed in approximately 2 GPU hours, highlighting its efficiency.
>
>
>
> **[Q3]**: While non-greedy sampling is an important setting for language generation, the paper only looks at greedy verification strategies. If possible it would be useful to include results for how ParaDecode performs with non-greedy sampling.
>
> **[A3]**: Thanks for the suggestion. Currently, we primarily focus on greedy decoding strategy as in line with previous works[3]. We also acknowledge the possibility of extending our approach to support non-greedy sampling strategies. For example, by adopting the rejection sampling algorithm introduced in speculative sampling [4,5] as our verification strategy, ParaDecode can maintain output parity with standard autoregressive decoding while supporting other non-greedy sampling methods such as temperature-based sampling.

---

> ### Author Response · Authors · 2024-11-29
> **Response to Reviewer XpyW [part 2/2]**
>
> **[Q4]**: Would you be able to speak more towards how ParaDecode achieves a speedup when different layers still need to be fetched from memory for each token being decoded in parallel. While I understand that the actual computation is batched, as each token has a corresponding unique weight operating on it, it seems as though the weight movement per token generated would not be decreased. Any clarification would be appreciated.
>
> **[A4]**: We would like to note that all parameter matrices of the model are already loaded into memory before the decoding process begins [1, 2, 3], so there is no need to repeatedly load parameters for each individual layer. Therefore, our batched matrix-vector multiplication does not incur additional overhead from loading weights into memory, compared to standard non-batched operations.
>
>
> While we acknowledge that ParaDecode does not specifically optimize the weight-fetching operation, the overhead of fetching weights from memory is minimal and does not significantly impact LLM decoding speed. Additionally, fetching weights is a fundamental operation in almost all computations, but optimizing this operation is typically orthogonal to the line of research focused on LLM decoding efficiency [1, 2, 3, 4, 5]. Therefore, the weight-fetching overhead in our method is comparable to that of other accelerated generation methods, such as speculative decoding [4, 5] or Medusa [6], which also adopt the standard weight movement operations.
>
>
> **[Q5]**: Tree based speculative decoding ([1], [2]]) sets the current state-of-the-art for speculative decoding techniques. Would tree based decoding work for ParaDecode?
>
> **[A5]**: We agree that tree-based speculative decoding has demonstrated promising performance. Typically, these methods first draft multiple candidate sequences **across several decoding steps** and then verify the sequences simultaneously, often requiring a draft model [2] or additional full LM heads [1]. In contrast, our method focuses on processing multiple tokens **within each decoding step** enabled by lightweight LM heads, achieving parallelism from an orthogonal perspective.
>
>
> We believe it is possible and interesting to explore how our method could potentially be combined with tree-based decoding techniques. For instance, one potential approach could be generating multiple tokens at certain intermediate layers using our lightweight LM heads, rather than introducing additional full LM heads as in [1] and always drafting tokens at the last layer. This strategy could allow for more efficient exploration of the decoding space while maintaining the benefits of both methods.
>
>
> **References**
> - [1] Cai et al. Medusa: Simple Framework for Accelerating LLM Generation with Multiple Decoding Heads. ICML 2024.
> - [2] Miao et al. SpecInfer: Accelerating Generative Large Language Model Serving with Tree-based Speculative Inference and Verification. ASPLOS 2024.
> - [3] Fu et al. Break the Sequential Dependency of LLM Inference Using Lookahead Decoding. ICML 2024.
> - [4] Leviathan et al. Fast Inference from Transformers via Speculative Decoding. ICML 2023.
> - [5] Chen et al. Accelerating Large Language Model Decoding with Speculative Sampling. arXiv:2302.01318
>
> Please let us know if you have any further questions, and we are happy to incorporate additional suggestions you might have! If you find our response satisfactory, we would be grateful if you could consider raising your score. Thanks again for your time and efforts!

---

> > ### Comment · Reviewer_XpyW · 2024-12-01
> >
> > Thank you to the authors for their clarifications. I still have a few questions regarding the work.
> >
> > > Fairness of SpecDecode baseline
> >
> > While the responses are generated by the model themselves, as stated in Appendix C the prompts are derived from the training splits of the evaluation datasets. My understanding is that the SpecDecode speculators are frozen and not further trained on the distribution of prompts in the evaluation. I think a minimal fair baseline would be to further train the speculators on the data the ParaDecode heads are trained on. My concern is that on the task which is most in-distribution for the CodeLlama speculator (HumanEval), the performance of SpecDecode is very similar to that of ParaDecode.
> >
> > > Weight movement with ParaDecode
> >
> > Sorry for not being more clear in my initial question, I understand that all parameters are loaded into GPU memory but my question is how ParaDecode can achieve a speedup in the memory-bandwidth bound regime that occurs for small batch size decoding. Namely, the bottleneck is moving model weights from off-chip memory to on-chip memory where the accelerator can then perform computations. All the works the authors cited (Medusa, SpecInfer, Lookahead, SpecDec) explicitly achieve a speedup by reducing the amount of memory movement per token generated. As ParaDecode does not reduce the memory movement from off-chip to on-chip as multiple layers need to be fetched at the same time, I am still unclear how ParaDecode achieves a speedup and think that the paper is incomplete without clarification.

---

> > > ### Author Response · Authors · 2024-12-04
> > > **Response to Reviewer XpyW [Round 2, Part 1/2]**
> > >
> > > We would like to express our gratitude to reviewer XpyW for taking the time to provide thoughtful feedback and a candid exchange of opinions about our work. We appreciate the reviewer’s insights and suggestions, which significantly helped us improve the presentation and positioning of our research.
> > >
> > > Although we realize that the discussion stage is closing soon and we may not be able to receive any further feedback from the reviewer, we'd like to take this final opportunity to address your comments and provide clarifications. Regardless of the final decision, we hope our responses demonstrate our appreciation for your insights and our commitment to improving our work. Thank you again for your efforts and for engaging with our submission.
> > >
> > > **[Round-2 Q1]**: Fairness of SpecDecode baseline. While the responses are generated by the model themselves, as stated in Appendix C the prompts are derived from the training splits of the evaluation datasets. My understanding is that the SpecDecode speculators are frozen and not further trained on the distribution of prompts in the evaluation. I think a minimal fair baseline would be to further train the speculators on the data the ParaDecode heads are trained on. My concern is that on the task which is most in-distribution for the CodeLlama speculator (HumanEval), the performance of SpecDecode is very similar to that of ParaDecode.
> > >
> > >
> > > **[Round-2 A1]**:  We appreciate the reviewer’s suggestion and acknowledge that further fine-tuning of the speculator on in-domain data may lead to improvements in speculative decoding. However, we would like to clarify that our setup is consistent with previous works such as Medusa, which also involves a separate training process for the additional LM heads while utilizing an off-the-shelf speculator.
> > >
> > > **[Round-2 Q2]**: Weight movement with ParaDecode. Sorry for not being more clear in my initial question, I understand that all parameters are loaded into GPU memory but my question is how ParaDecode can achieve a speedup in the memory-bandwidth bound regime that occurs for small batch size decoding. Namely, the bottleneck is moving model weights from off-chip memory to on-chip memory where the accelerator can then perform computations. All the works the authors cited (Medusa, SpecInfer, Lookahead, SpecDec) explicitly achieve a speedup by reducing the amount of memory movement per token generated. As ParaDecode does not reduce the memory movement from off-chip to on-chip as multiple layers need to be fetched at the same time, I am still unclear how ParaDecode achieves a speedup and think that the paper is incomplete without clarification.
> > >
> > > **[Round-2 A2]**: We acknowledge that batching and speculative decoding are two orthogonal approaches to improve GPU utilization in LLM inference [1]. To further justify the speedups achieved by our method, we would like to provide the following clarifications.
> > >
> > > ## Memory access latency can be reduced to a comparable level of compute latency
> > >
> > > - **Memory access latency is hardware-dependent and drastically reduced in the latest GPUs**: It is worth noting that early works [2] concluded that memory latency was much greater than computer latency on less performant GPUs such as A5000, which have limited memory bandwidth and smaller on-chip caches. Recent GPUs such as the H100/H200 have featured **increased memory bandwidth** (e.g., H100 is 5x faster than the A5000 in memory access), **larger on-chip caches** (e.g., H100’s L2 cache is 12.5x larger than the A5000’s), and **improved data transfer protocols** (e.g., H100 can prefetch data into cache while executing computations), all of which significantly reduce memory access latency. In our work, we conduct experiments on H100 GPUs with much higher bandwidth, making it possible for ParaDecode to achieve meaningful speedups by increasing computational throughput via the batching technique. This suggests the possibility that **memory access may no longer be the sole or primary bottleneck in LLM inference on high-end GPU architectures**.
> > >
> > > - **Memory access operations can be further optimized via software-level mechanisms**: In addition to hardware improvements, recent advances have introduced multiple techniques to further reduce memory overhead, such as swapping [3], recomputation [4], and their combination [5]. Notable works include vLLM [6], which introduces a novel memory management technique called PageAttention to optimize memory access operations. FlashAttention [7] applies tiling and kernel optimizations to reduce the peak memory required for attention computations and minimize memory I/O costs.

---

> > > ### Author Response · Authors · 2024-12-04
> > > **Response to Reviewer XpyW [Round 2, Part 2/2]**
> > >
> > > ## Why can ParaDecode achieve meaningful speedups?
> > > While the above advances have significantly reduced memory access latency, we acknowledge that LLM inference can remain memory-bandwidth bound given its limited arithmetic intensity [2]. This allows ParaDecode to further increase computational throughput by parallelizing computation across multiple tokens via batching, maximizing the utilization of GPU tensor cores and reducing the compute latency. Moreover, the batched operations also save other overheads such as kernel launches and function calls, as sequential operations typically involve such repeated non-computational latency.
> > > Despite being memory-bandwidth bound, we believe the non-trivial speedups achieved by ParaDecode could potentially serve as an empirical evidence for the argument that **memory latency is significantly reduced to a level comparable to or even less impactful than compute latency on modern GPUs**, as suggested by the aforementioned GPU advancements. This phenomenon is likely to become more evident in the future, as we can expect compute latency in LLMs to further increase with models generating longer sequences. For example, recent state-of-the-art LLMs such as OpenAI’s o1, tend to produce long chains of thought for complex reasoning.
> > >
> > > Indeed, this implication is consistent with previous findings [8], which also observed that compute latency can be much larger than memory latency during LLM inference:
> > > > “However, as the input batch size and the sequence length increase, the compute-bound factor becomes predominant, overshadowing the influence of memory-bound. In such cases, matrix multiplication consumes up to 80% of the inference time, as reported by LightSeq [9].”
> > > ## Summary
> > > To summarize, our finding suggests that even LLM inference is theoretically memory-bound, in practice **reducing compute and memory latency could be both important** because of the complicated interplay between computation and memory access operations. For example, in the LLM forward passes, we need to frequently switch between memory fetching and matrix multiplications, leading to imperfect overlap of computation and memory access that jointly affects the overall inference efficiency. In other words, provided that arithmetic operations are potentially playing an increasingly critical role in overall inference efficiency on modern GPUs, it is entirely possible that optimizing compute latency can yield significant practical improvements in throughput. We will include more in-depth experiments to analyze the latency of memory access versus arithmetic operations in our next revision.
> > >
> > > **References**
> > > - [1] Su et al. The Synergy of Speculative Decoding and Batching in Serving Large Language Models. arXiv:2310.18813
> > > - [2] Kim et al. SqueezeLLM: Dense-and-Sparse Quantization. ICML 2024.
> > > - [3] Huang et al. SwapAdvisor: Pushing Deep Learning Beyond the GPU Memory Limit via Smart Swapping. ASPLOS 2020.
> > > - [4] Jain et al. Checkmate: Breaking the Memory Wall with Optimal Tensor Rematerialization. MLSys 2020.
> > > - [5] Patil et al. POET: Training Neural Networks on Tiny Devices with Integrated Rematerialization and Paging. ICML 2022.
> > > - [6] Kwon et al. Efficient Memory Management for Large Language Model Serving with PagedAttention. SOSP 2023
> > > - [7] Dao et al. FlashAttention: Fast and Memory-Efficient Exact Attention with IO-Awareness. NeurIPS 2022
> > > - [8] Zhang et al. FlattenQuant: Breaking Through the Inference Compute-bound for Large Language Models with Per-tensor Quantization. COLING 2024.
> > > - [9] Wang et al. LightSeq: A High Performance Inference Library for Transformers. NAACL 2021.

---

### Official Review · Reviewer_ebtN · 2024-11-04

**Soundness:** 3
**Presentation:** 3
**Contribution:** 3
**Rating:** 5
**Confidence:** 5

**Summary:**

Autoregressive decoding method without auxiliary model, based on the observation that simple tokens can be predicted using intermediate layer representations. Paradecode generate tokens at the intermediate layer to allow next token generation immediately. The computation of next token generation and current token computation can be parallelized with batched matrix multiplication.

**Strengths:**

* Paper is well written and easy to follow.
* Combining layer skipping and speculative decoding is an interesting idea and makes sense. Both approaches compliment each other.
* Using per-layer LM head seems a right choice to generate a next token based on intermediate with high fidelity.

**Weaknesses:**

* Missing experiments to claim the generalization of lightwight layerwise LM head. Also it will increase the computation overhead by adding additional parameters. How does latency change with the LM heads?
* Latency is critical in LM inference and paper doesn’t include any latency or throughput experiments.

**Questions:**

* Is this method compatible with continuous batching? continuous batching is standard of LLM inference so i want to know how this method can seamlessly integrated with standard inference systems.
* What is the overhead from LM Head at each intermediate layer?
* How to determine Confidence threshold? It says it’s a hyperparameter but I wonder how it can be set in practice.
* How does the confidence threshold per layer differ? for instance I expect confidence threshold of early layers should be very high because it haven’t passed many layers yet. Also could you provide the average statistics of when the next token generation could start by layers?  (e.g., X tokens started next token gen after layer 0, Y tokens started next token gen after layer 1, ..)
* Is BMM applicable when weights are different? (across layer, weights are different but is it possible?) As far as I understand, bmm only works for batch dimension, so for input of size (b,n,m) and weight (b,m,p), output becomes (b,n,p). if weight is different for each input, isn’t it just concatenated matrix multiplication?

---

> ### Author Response · Authors · 2024-11-29
> **Response to Reviewer ebtN [part 1/2]**
>
> We sincerely thank the reviewer for the constructive comments, and we’ve added additional experimental results per the reviewer’s suggestions.
>
> **[Q1]**: Missing experiments to claim the generalization of lightwight layerwise LM head. Also it will increase the computation overhead by adding additional parameters. How does latency change with the LM heads?
>
> **[A1]**: We’d like to clarify that the additional overhead introduced by our lightweight LM head is minimal, as the lightweight LM (with dimension hidden_dim × vocab_size) head simply maps the intermediate-layer hidden states to the final-layer hidden states, which are then processed by the original LM head. This design ensures efficiency without significantly compromising on performance.
>
> Per the reviewer’s suggestion, we conducted an additional experiment by training a full LM head (with dimensions hidden_dim × vocab_size) on the Llama-3.1-8B-Instruct model. The results demonstrate that its performance closely matches our proposed lightweight LM head, achieving comparable speedups (86.41 token/s vs. 85.79 token/s on GSM8K). These findings underscore the effectiveness of our method while maintaining minimal computational overhead.
>
> **[Q2]**: Latency is critical in LM inference and paper doesn’t include any latency or throughput experiments.
>
> **[A2]**: Thanks for the suggestions. Below we present the latency analysis across three benchmarks:
>
> |Model Size|Method| XSUM | HumanEval | GSM8K
> |:-:|:-:|:-:|:-:|:-:|
> |8B|Baseline| 61.37 token/s| 61.03 token/s | 60.57 token/s
> |8B|ParaDeocde| **70.76** token/s | **80.04** token/s|  **85.79** token/ s
> |34B|Baseline| 27.70 token/s | 29.81 token/s | 30.35 token/s
> |34B|ParaDeocde| **34.25** token/s | **43.22** token/s | **46.28** token/s
>
> These results, which align with the speedups shown in **Table 1**, demonstrate that our ParaDecode method consistently enhances throughput compared to standard autoregressive decoding across various tasks and model sizes.
>
>
> **[Q3]**: Is this method compatible with continuous batching? continuous batching is standard of LLM inference so i want to know how this method can seamlessly integrated with standard inference systems.
>
> **[A3]**: Thanks for bringing this up. We agree that continuous batching is a useful mechanism for LLM serving, allowing multiple inference requests to be processed in parallel. Indeed, our method is compatible with continuous batching, as it processes tokens in parallel using the batching mechanism, which is native to existing LLM inference frameworks. For multiple decoding sequences, if each sequence invokes multiple parallel token processes, we can simply expand the batch size accordingly to process multiple tokens across multiple sequences.
>
> **[Q4]**: What is the overhead from LM Head at each intermediate layer?
>
> **[A4]**:  We’d like to clarify that the additional overhead introduced by our lightweight LM head is minimal, as the lightweight LM (with dimension $hidden\\_dim × vocab\\_size$) head simply maps the intermediate-layer hidden states to the final-layer hidden states, which are then processed by the original LM head.
> - **Memory overhead**: Specifically, as stated in Section 2.1 (Line 202), our method only requires $4096 \times 4096 = 16$M parameters for the lightweight LM head, which can be fine-tuned within approximately 2 GPU hours.
> - **Latency overhead**: Given the lightweight nature of our intermediate LM head, its latency overhead is actually negligible – To demonstrate, we evaluated this operation using the 8B model on the XSUM dataset, and the latency is only 0.4 ms/token, which is trivial compared to the full pass of decoding a token that takes 14.13 ms/token (i.e., 70.76 tokens/s, as stated in our response to Q2).

---

> ### Author Response · Authors · 2024-11-29
> **Response to Reviewer ebtN [part 2/2]**
>
> **[Q5]**: How to determine the confidence threshold? It says it’s a hyperparameter but I wonder how it can be set in practice.
>
> **[A5]**:  The confidence threshold is a hyperparameter that helps balance the frequency of early predictions and their success rate, ultimately improving the overall speedup. In our implementation, we only use a single lightweight LM head positioned at the middle layer (e.g., 16-th layer for a model with 32 layers) of the model. This design choice is based on our observation that intermediate LM heads inserted at very shallow layers (e.g., layer 1 or 2) struggle to provide reasonable token predictions due to insufficient contextual information, while those inserted at very deep layers offer minimal speedup benefits as they are too close to the final layer. Therefore, we opt to insert only one additional LM head at the middle layer to balance prediction accuracy and computational efficiency.
>
>
> As illustrated in **Figure 3(b)**, the lightweight LM head at the middle layer can confidently make reliable early predictions. Therefore, we can safely set the threshold very low or even to zero to allow early predictions without significantly compromising the success rate. Moreover, the choice of threshold does not impact the generation performance, as we incorporate a verification step that ensures the output matches the result of standard decoding, regardless of the threshold setting.
>
> **[Q6]**: How does the confidence threshold per layer differ? for instance I expect confidence threshold of early layers should be very high because it haven’t passed many layers yet. Also could you provide the average statistics of when the next token generation could start by layers? (e.g., X tokens started next token gen after layer 0, Y tokens started next token gen after layer 1, ..)
>
> **[A6]**: Thanks for the question. We agree that the confidence threshold should be set higher for earlier layers to better balance the frequency of early predictions and their success rate. However, as mentioned in our response to Q5, we implement our method using a single lightweight LM head positioned at the middle layer (e.g., the 16th layer for a model with 32 layers). As demonstrated in **Figure 3(b)**, this configuration already delivers promising early predictions, and thus we can use a lower threshold (or even zero) to encourage early predictions and achieve significant speedups.
>
> **[Q7]**: Is BMM applicable when weights are different? (across layer, weights are different but is it possible?) As far as I understand, bmm only works for batch dimension, so for input of size (b,n,m) and weight (b,m,p), output becomes (b,n,p). if weight is different for each input, isn’t it just concatenated matrix multiplication?
>
> **[A7]**: Yes, your understanding of the standard batch matrix multiplication (BMM) operation is correct, which typically works with consistent weights across the batch. However, in our case, we will first concatenate the weights from different layers before applying matrix multiplication with the batched inputs for multiple tokens. Specifically, the batch dimension represents the number of tokens being processed in parallel.
>
> For example, in a batched projection operation (e.g., q_proj/k_proj/v_proj), we use a batched weight with the shape (bsz, dim_in, dim_out) and a batched input with the shape (bsz, seq_len, dim), where dim = dim_in, and seq_len = 1 as KV cache is enabled and we only need to process the current token. The resulting output will have the shape (bsz, seq_len, dim_out), denoting the projected representation. This approach allows us to handle the weights from different layers while still leveraging the efficiency benefits of batch matrix multiplication.
>
> Please let us know if you have any further questions, and we are happy to incorporate additional suggestions you might have! If you find our response satisfactory, we would be grateful if you could consider raising your score. Thanks again for your time and efforts!

---

### Official Review · Reviewer_o7wq · 2024-11-05

**Soundness:** 1
**Presentation:** 3
**Contribution:** 2
**Rating:** 3
**Confidence:** 4

**Summary:**

This paper proposes ParaDecode, a new approach for speeding up LLM decoding by processing several tokens in parallel. It does so by: (1) adding lightweight LM heads at intermediate layers in an LLM, (2) While processing token $t$, if one of the intermediate LM heads is confident enough in token $t+1$, it samples that token and begins processing it at the bottom of the network, while continuing to process token $t$ with the later layers of the network, (3) finally, once token $t$ has been fully processed by the network, ParaDecode samples the next token $t'+1$, and if it disagrees with token $t+1$, it discards the partial computation of $t+1$ and begins processing $t'+1$. In this way, the processing of several tokens can be overlapped, thus speeding up decoding. Empirically, ParaDecode demonstrates speedups of up to 1.53x relative to autoregressive decoding.

**Strengths:**

- ParaDecode is a novel algorithm for speedup up LLM decoding by processing several tokens in parallel.
- The approach of training lightweight LM heads $E^{(i)}$ by just training a rotation matrix on the model's frozen LM is nice. $E^{(i)} = E^* T^{(i)}$, where $E^*$ is the models pre-trained LM head. This way, only $d^2$ parameters must be trained per intermediate LM head, instead of $d * vocab$.

**Weaknesses:**

- **Most importantly**: I do not see why this method gives speedups relative to autoregressive decoding.  In particular, I'm confused by why the batched matrix-vector multiply is able to give meaningful speedups relative to sequential autoregressive decoding. It seems the time to execute this batched operation should essentially be equal to the time to load all the parameter matrices into memory (because it is a memory-bound operation); thus it's unclear to me why batching this operation is faster than performing these operations sequentially (which is what normal autoregressive decoding does).
- The baselines seem quite weak. Speculative decoding can generally give 1.5x-3x speedups (e.g., using Llama-3.1-8B as a speculator for Llama-3.1-70B). The choice of draft/target model combinations is a bit unusual (CodeLlama, Llama 2)---why not just use Llama 3.1 and 3.2 models, which come in many sizes?
- The paper claims ParaDecode is easier to get working than speculative decoding when a small draft model is not readily available. However, the intermediate LM heads must still be trained. Therefore, it's unclear to me why this training is simpler than, for example, the LM head training in Medusa. Medusa gives meaningfully larger speedups than those presented in this paper.
- This paper bears meaningful resemblance to the speculative decoding algorithm in LayerSkip, which is not adequately discussed or compared to in my opinion (although in LayerSkip the whole model is trained, a meaningful difference).

**Questions:**

- See my first bullet in the weaknesses section: Why is batched matrix-vector multiply faster than sequential matrix-vector multiplies?
- Can you explain the consistency results in more detail (Table 2)? Are these with greedy decoding? I'm confused by the results being so low.
- Is there an intermediate LM head at every single layer? If so, I'm confused by how using the threshold $\gamma = 0$ can work, as this would mean all tokens early exit after layer 1, which seems like it would perform quite poorly (many rejected tokens).
- In Figure 5, what are the thresholds that are references in the legends of the figures? Speculative decoding does not have any thresholds that I'm aware of.
- What batch sizes are used in the experiments? Is it always batch size of 1 during ParaDecode inference?
- NIT: On line 247, shouldn't it be $l^{(t_1)} > l^{(t_2)} > \ldots$?

---

> ### Author Response · Authors · 2024-11-29
> **Response to Reviewer o7wq [part 1/3]**
>
> We sincerely thank the reviewer for the constructive suggestions.
>
> **[Q1]**: I'm confused by why the batched matrix-vector multiply is able to give meaningful speedups relative to sequential autoregressive decoding. It seems the time to execute this batched operation should essentially be equal to the time to load all the parameter matrices into memory (because it is a memory-bound operation); thus it's unclear to me why batching this operation is faster than performing these operations sequentially (which is what normal autoregressive decoding does). Why is batched matrix-vector multiply faster than sequential matrix-vector multiplies?
>
> **[A1]**: Thanks for raising this point. It seems there might be some confusion regarding our problem setting, and we would like to make the following clarifications:
>
> **Decoding multiple requests**: We acknowledge that the reviewer’s concern is particularly relevant to the case of decoding only a single request, where loading model weights can dominate the runtime over the actual computation. However, our work aims at the general scenario of decoding multiple requests, as consistent with previous LLM decoding acceleration works [1,2,3]. In this setting, the model weights are loaded into memory once and do not require reloading for each subsequent request.
>
> **Runtime evaluation**: In this line of research, the runtime evaluation typically excludes the time taken to load model weights since these weights can be loaded once and cached for efficient reuse. In other words, we only count the time elapsed during the operation of `model.generate()` for each request (e.g., as illustrated in [Self-SpecDecode’s codebase](https://github.com/dilab-zju/self-speculative-decoding/blob/main/decoding.py#L356-L362)). This allows us to accurately assess the efficiency of the decoding algorithm itself, without any distractions from other operations.
>
> Therefore, once the model weights are loaded, they will remain in memory, thus avoiding repeated loading overhead and enabling our method to achieve meaningful speedups – standard autoregressive decoding processes each token sequentially, failing to fully leverage the parallelism of GPU accelerators. In contrast, our method leverages batched matrix-vector multiplication to process multiple tokens in parallel, significantly improving the efficiency of the decoding process.
>
>
>
> **[Q2]**: The baselines seem quite weak. Speculative decoding can generally give 1.5x-3x speedups (e.g., using Llama-3.1-8B as a speculator for Llama-3.1-70B). The choice of draft/target model combinations is a bit unusual (CodeLlama, Llama 2)---why not just use Llama 3.1 and 3.2 models, which come in many sizes?
>
> **[A2]**: We appreciate the reviewer’s suggestion to test baselines with more recent models. However, we would like to clarify that our choice of drafter/verifier models directly follows the setups of previous works [4,5] for the following reasons:
>
> **Compatibility with previous works**: Given the rapid releases of LLMs, it’s challenging to establish completely fair and comprehensive comparisons with past works. For instance, previous works like [4] rely on model-specific configurations (e.g., the skipped intermediate layers for each model are pre-configured before the decoding process). Additionally, their open-source codebase was highly intertwined with obsolete versions of libraries that only supported CodeLlama and Llama 2 (e.g., the authors recommended using transformers version v4.33.1 to ensure compatibility with their codebase in their GitHub [issue-8](https://github.com/dilab-zju/self-speculative-decoding/issues/8#issuecomment-1860190005) and [issue-14](https://github.com/dilab-zju/self-speculative-decoding/issues/14#issuecomment-1996249908), while the support for Llama 3.1 was not introduced until transformers version v4.43.0).
> These practical limitations restrict the direct applicability of their codebase to newer models due to:
> - (1) The lack of implementation of essential functionalities (e.g., grouped-query attention, RoPE scaling), which necessitates substantial development to the codebase to support newer models.
> - (2) The requirement of an extensive Bayesian Optimization search to determine the configurations for specific models, as suggested in this [GitHub issue](https://github.com/dilab-zju/self-speculative-decoding/issues/20).
>
>
> **Resource constraints**: We agree that speculative decoding may achieve higher speedups by using larger target models, however employing an 8B speculator with a 70B target model requires loading both models into memory simultaneously. Unfortunately, we do not have the resources to accommodate such a setup. For example, this combination would require 156 GB of GPU memory (even in half-precision), which surpasses the capacity of most high-end GPUs (e.g., an A100/H100 only has 80GB memory). Therefore, we opt to experiment with up to 34B models, as in line with previous works [1,2,5].

---

> > ### Comment · Reviewer_o7wq · 2024-12-02
> >
> > I thank the authors for their responses, but leave my review unchanged.  Like reviewer vTw5 (see exchange about their Q3), I believe there is an issue with how the authors are discussing “loading” the parameters. As reviewer vTw5 noted, the bottleneck during decoding is moving model parameters from “global” (HBM) to “local” storage where the computations are actually performed. Because of this, it is unclear how the proposed algorithm can yield any speedups.

---

> > > ### Author Response · Authors · 2024-12-04
> > > **Response to Reviewer o7wq [Round 2, Part 1/2]**
> > >
> > > We would like to express our gratitude to reviewer o7wq for taking the time to provide thoughtful feedback and a candid exchange of opinions about our work. We appreciate the reviewer’s insights and suggestions, which significantly helped us improve the presentation and positioning of our research.
> > >
> > > Although we realize that the discussion stage is closing soon and we may not be able to receive any further feedback from the reviewer, we'd like to take this final opportunity to address your comments and provide clarifications. Regardless of the final decision, we hope our responses demonstrate our appreciation for your insights and our commitment to improving our work. Thank you again for your efforts and for engaging with our submission.
> > >
> > > **[Round-2 Q1]**:I thank the authors for their responses, but leave my review unchanged. Like reviewer vTw5 (see exchange about their Q3), I believe there is an issue with how the authors are discussing “loading” the parameters. As reviewer vTw5 noted, the bottleneck during decoding is moving model parameters from “global” (HBM) to “local” storage where the computations are actually performed. Because of this, it is unclear how the proposed algorithm can yield any speedups.
> > >
> > > **[Round-2 A1]**: We acknowledge that batching and speculative decoding are two orthogonal approaches to improve GPU utilization in LLM inference [1]. To further justify the speedups achieved by our method, we would like to provide the following clarifications.
> > >
> > > ## Memory access latency can be reduced to a comparable level of compute latency
> > >
> > > - **Memory access latency is hardware-dependent and drastically reduced in the latest GPUs**: It is worth noting that early works [2] concluded that memory latency was much greater than computer latency on less performant GPUs such as A5000, which have limited memory bandwidth and smaller on-chip caches. Recent GPUs such as the H100/H200 have featured **increased memory bandwidth** (e.g., H100 is 5x faster than the A5000 in memory access), **larger on-chip caches** (e.g., H100’s L2 cache is 12.5x larger than the A5000’s), and **improved data transfer protocols** (e.g., H100 can prefetch data into cache while executing computations), all of which significantly reduce memory access latency. In our work, we conduct experiments on H100 GPUs with much higher bandwidth, making it possible for ParaDecode to achieve meaningful speedups by increasing computational throughput via the batching technique. This suggests the possibility that **memory access may no longer be the sole or primary bottleneck in LLM inference on high-end GPU architectures**.
> > >
> > > - **Memory access operations can be further optimized via software-level mechanisms**: In addition to hardware improvements, recent advances have introduced multiple techniques to further reduce memory overhead, such as swapping [3], recomputation [4], and their combination [5]. Notable works include vLLM [6], which introduces a novel memory management technique called PageAttention to optimize memory access operations. FlashAttention [7] applies tiling and kernel optimizations to reduce the peak memory required for attention computations and minimize memory I/O costs.

---

> > > ### Author Response · Authors · 2024-12-04
> > > **Response to Reviewer o7wq [Round 2, Part 2/2]**
> > >
> > > ## Why can ParaDecode achieve meaningful speedups?
> > >
> > > While the above advances have significantly reduced memory access latency, we acknowledge that LLM inference can remain memory-bandwidth bound given its limited arithmetic intensity [2]. This allows ParaDecode to further increase computational throughput by parallelizing computation across multiple tokens via batching, maximizing the utilization of GPU tensor cores and reducing the compute latency. Moreover, the batched operations also save other overheads such as kernel launches and function calls, as sequential operations typically involve such repeated non-computational latency.
> > >
> > > Despite being memory-bandwidth bound, we believe the non-trivial speedups achieved by ParaDecode could potentially serve as an empirical evidence for the argument that **memory latency is significantly reduced to a level comparable to or even less impactful than compute latency on modern GPUs**, as suggested by the aforementioned GPU advancements. This phenomenon is likely to become more evident in the future, as we can expect compute latency in LLMs to further increase with models generating longer sequences. For example, recent state-of-the-art LLMs such as OpenAI’s o1, tend to produce long chains of thought for complex reasoning.
> > >
> > > Indeed, this implication is consistent with previous findings [8], which also observed that compute latency can be much larger than memory latency during LLM inference:
> > > > “However, as the input batch size and the sequence length increase, the compute-bound factor becomes predominant, overshadowing the influence of memory-bound. In such cases, matrix multiplication consumes up to 80% of the inference time, as reported by LightSeq [9].”
> > >
> > > ## Summary
> > > To summarize, our finding suggests that even LLM inference is theoretically memory-bound, in practice **reducing compute and memory latency could be both important** because of the complicated interplay between computation and memory access operations. For example, in the LLM forward passes, we need to frequently switch between memory fetching and matrix multiplications, leading to imperfect overlap of computation and memory access that jointly affects the overall inference efficiency. In other words, provided that arithmetic operations are potentially playing an increasingly critical role in overall inference efficiency on modern GPUs, it is entirely possible that optimizing compute latency can yield significant practical improvements in throughput. We will include more in-depth experiments to analyze the latency of memory access versus arithmetic operations in our next revision.
> > >
> > > **References**
> > > - [1] Su et al. The Synergy of Speculative Decoding and Batching in Serving Large Language Models. arXiv:2310.18813
> > > - [2] Kim et al. SqueezeLLM: Dense-and-Sparse Quantization. ICML 2024.
> > > - [3] Huang et al. SwapAdvisor: Pushing Deep Learning Beyond the GPU Memory Limit via Smart Swapping. ASPLOS 2020.
> > > - [4] Jain et al. Checkmate: Breaking the Memory Wall with Optimal Tensor Rematerialization. MLSys 2020.
> > > - [5] Patil et al. POET: Training Neural Networks on Tiny Devices with Integrated Rematerialization and Paging. ICML 2022.
> > > - [6] Kwon et al. Efficient Memory Management for Large Language Model Serving with PagedAttention. SOSP 2023
> > > - [7] Dao et al. FlashAttention: Fast and Memory-Efficient Exact Attention with IO-Awareness. NeurIPS 2022
> > > - [8] Zhang et al. FlattenQuant: Breaking Through the Inference Compute-bound for Large Language Models with Per-tensor Quantization. COLING 2024.
> > > - [9] Wang et al. LightSeq: A High Performance Inference Library for Transformers. NAACL 2021.

---

> ### Author Response · Authors · 2024-11-29
> **Response to Reviewer o7wq [part 2/3]**
>
> **[Q3]**: The paper claims ParaDecode is easier to get working than speculative decoding when a small draft model is not readily available. However, the intermediate LM heads must still be trained. Therefore, it's unclear to me why this training is simpler than, for example, the LM head training in Medusa.
>
> **[A3]**: While it is true that our method requires training intermediate lightweight LM heads, the key difference lies in the scale and complexity of these heads. Previous works like Medusa require multiple large LM heads (e.g., 5 heads as suggested in [1]), each being a matrix of size $d \times vocab\\_{size}$. For example, for a Llama-3-8B-based Medusa instance with 5 heads, the additional parameters can be up to $5 \times 4096 \times 128256 = 2.63$B parameters, which is approximately 156 times larger than the parameter count in our method. As stated in **Section 2.1 (Line 202)**, our method only requires $4096 \times 4096 = 16$M parameters for the lightweight LM head, making it much more efficient.
>
>
> Moreover, training Medusa’s LM heads requires at least 5 GPU hours for the basic implementation (Medusa-1) [1], and the full Medusa model (Medusa-2) involves fine-tuning the entire model along with the Medusa heads trained in the first stage, resulting in even larger training overheads (e.g., the two-stage training can take 20 GPU hours). In contrast, training the lightweight LM head in our ParaDecode can be completed in approximately 2 GPU hours, highlighting its efficiency.
>
> **[Q4]**: This paper bears meaningful resemblance to the speculative decoding algorithm in LayerSkip, which is not adequately discussed or compared to in my opinion (although in LayerSkip the whole model is trained, a meaningful difference).
>
> **[A4]**: Thank you for your suggestion, we have added discussions on LayerSkip in Section 5.2. However, we believe there are key differences that make our approach more practical and efficient:
>
> - **Simple training objective**: LayerSkip requires a complex design of training strategies, such as layer dropout and early exit loss, whereas our method uses a straightforward KL divergence loss.
> - **Model-agnostic speedup strategy**: LayerSkip achieves speedup by implementing a cache reuse mechanism tailored for the Llama model structure. In contrast, our speedup mechanism is model-agnostic, relying solely on the native batching technique, alleviating the need for model-specific designs.
> - **Efficiency and parity**: Our method only needs to train a small number of parameters (e.g., 16M), making it lightweight and efficient, without the need to fine-tune the entire model as in LayerSkip. Additionally, LayerSkip does not guarantee output parity with standard autoregressive decoding, which makes it not directly comparable to our method.
>
> **[Q5]**: Can you explain the consistency results in more detail (Table 2)? Are these with greedy decoding? I'm confused by the results being so low.
>
> **[A5]**: Yes, the results in Table 2 are based on greedy decoding, and we used strict string matching to determine whether the outputs of the accelerated decoding strategies are consistent with those from standard greedy decoding.
>
> We would like to note that, in practice, the output of speculative decoding can differ slightly from standard decoding due to numerical precision inaccuracies and minor variations in token probabilities during computation (please refer to this [GitHub issue](https://github.com/huggingface/transformers/issues/30413)). In most cases, the consistency of speculative decoding is high. However, for coding tasks, the consistency is comparatively lower. This is because coding tasks are more sensitive to even trivial differences, such as variable naming, code formatting, or syntactic structure, which can lead to failures in strict consistency matching.

---

> ### Author Response · Authors · 2024-11-29
> **Response to Reviewer o7wq [part 3/3]**
>
> **[Q6]**: Is there an intermediate LM head at every single layer? If so, I'm confused by how using the threshold \gamma=0 can work, as this would mean all tokens early exit after layer 1, which seems like it would perform quite poorly (many rejected tokens).
>
>
> **[A6]**: No, in our experiments, we introduce only one additional lightweight LM head at the middle layer ($L/2$), where $L$ is the total number of layers in the model. This design choice is based on our observation that intermediate LM heads inserted at very shallow layers (e.g., layer 1 or 2) struggle to provide reasonable token predictions due to insufficient contextual information, while those inserted at very deep layers offer minimal speedup benefits as they are too close to the final layer. Therefore, we opt to insert only one additional LM head at the middle layer to balance prediction accuracy and computational efficiency.
>
> As shown in **Figure 3(b)**, for a Llama-3.1-8B-Instruct model with 32 layers, early predictions made at the 16th layer using the lightweight LM head can confidently achieve almost identical probabilities as those from the last layer, ensuring the accuracy of early predictions. Therefore, even with a threshold of $\gamma = 0$, the model can still generate meaningful early predictions from this middle layer.
>
> **[Q7]**: In Figure 5, what are the thresholds that are references in the legends of the figures? Speculative decoding does not have any thresholds that I'm aware of.
>
> **[A7]**: The thresholds in Figure 5 correspond to the strategy for dynamically adjusting the number of draft tokens, as implemented in the Transformers library. Specifically, if the drafter model’s confidence in the current token prediction falls below a predefined threshold, the token generation process will immediately stop at that iteration, even if the maximum number of speculative tokens has not been reached. As highlighted in the [HuggingFace blog post](https://huggingface.co/blog/dynamic_speculation_lookahead), this dynamic approach leads to improved speedup compared to vanilla speculative decoding that generates a fixed number of draft tokens.
>
> **[Q8]**: What batch sizes are used in the experiments? Is it always batch size of 1 during ParaDecode inference?
>
> **[A8]**:  Yes, following the standard settings in previous works [1, 4], our experiments primarily focus on scenarios with a batch size of one, which is representative of common use cases where LLMs are locally hosted for personal use.
>
> **[Q9]**: NIT: On line 247, shouldn't it be $l^{t_1} \gt l^{t_2} \gt \dots$?
>
> **[A9]**:  Thanks for the careful review! We’ve corrected this typo in our updated manuscript.
>
> **References**
> - [1] Cai et al. Medusa: Simple Framework for Accelerating LLM Generation with Multiple Decoding Heads. ICML 2024.
> - [2] Leviathan et al. Fast Inference from Transformers via Speculative Decoding. ICML 2023.
> - [3] Miao et al. SpecInfer: Accelerating Generative Large Language Model Serving with Tree-based Speculative Inference and Verification. ASPLOS 2024.
> - [4] Zhang et al. Draft & Verify: Lossless Large Language Model Acceleration via Self-Speculative Decoding. ACL 2024.
> - [5] Elhoushi et al. LayerSkip: Enabling Early Exit Inference and Self-Speculative Decoding. ACL 2024.
>
> Please let us know if you have any further questions, and we are happy to incorporate additional suggestions you might have! If you find our response satisfactory, we would be grateful if you could consider raising your score. Thanks again for your time and efforts!

---

### Meta-Review · Area_Chair_KPNd · 2024-12-23

**Metareview:**

ParaDecode proposes combining layer-skipping-based decoding with parallel cache computation to accelerate LLM token generation. In ParaDecoding, light-weight LM heads are attached to intermediate model layers. When LM heads are sufficiently confident in a next token prediction, ParaDecode takes that token as the output and continues on to start decoding the next token. ParaDecode also employs a verification scheme where the early predicted token’s identity is verified against the token that would have been predicted had decoding not exited early.

However, some of the key claims made in this paper is not solid, e.g., several reviewers pointed out that LM heads still need to be tuned, even though the authors position one of ParaDecode's key advantage is overcoming the training of smaller, draft models. Also, some of the comparisons in the evaluation are problematic.

**Additional Comments On Reviewer Discussion:**

The authors provided rebuttals to address some reviewer's questions, but most reviewers remained negative about this paper.

---

### Decision · Program_Chairs · 2025-01-22

Reject